

# VEIN v0.2.2: an R package for bottom-up Vehicular Emissions Inventories

Sergio Ibarra-Espinosa[1], Rita Ynoue[1], Shane O'Sullivan[2], Edzer Pebesma[3], Maria de Fátima Andrade[1], and Mauricio Osses[4]

[1]Department of Atmospheric Sciences, Universidade de São Paulo, Rua do Matão 1226, São Paulo, SP, Brazil
[2]Department of Pathology, Faculdade de Medicina, Universidade de São Paulo, Av. Dr. Arnaldo 455, São Paulo, SP, Brazil
[3]Institute for Geoinformatics, Westfälische Wilhelms-Universität Münster, Heisenbergstraße 2, 48149 Münster, Germany
[4]Department of Mechanical Engineering, Universidad Técnica Federico Santa María, Vicuña Mackenna 3939, Santiago, Chile

*Correspondence to:* Sergio Ibarra-Espinosa (zergioibarra@gmail.com)

**Abstract.**

Emission inventories are the quantification of pollutants from different sources. They provide important information not only for climate and weather studies, but also for urban planning and environmental health protection. We developed an open source model (named VEIN v0.2.2) that provides high resolution vehicular emissions inventories for different fields of studies. We

focused on vehicular sources at street and hourly levels due to the current lack of information about these sources, mainly in developing countries. The type of emissions covered by VEIN are: exhaust (hot and cold) and evaporative considering the deterioration of the factors. VEIN also performs speciation and incorporates functions to generate and spatially allocate emissions databases. It allows users to load their own emissions factors, but it also provides emissions factors from the road transport model (Copert), the United States Environmental Protection Agency (EPA) and Brazilian databases. The VEIN model

reads, distributes by age of use and extrapolates hourly traffic data, and estimates hourly and spatially emissions. Based on our knowledge, VEIN is the first bottom-up vehicle emissions software that allows input to the WRF-Chem model. Therefore, the VEIN model provides an important, easy and fast way of elaborating or analyzing vehicular emissions inventories, under different scenarios. The VEIN results can be used as an input for atmospheric models, health studies, air quality standardizations and decision making.

## 1 Introduction

Emissions inventory is a quantification of pollutants discharged into the atmosphere by different sources (Pulles and Heslinga, 2010). This quantification is vital for regulatory and scientific purposes, because it allows to monitor the state of the Earth's atmosphere and climate. It also allows to create air quality standards, which will protect ecosystems and human health. For instance, the Intergovernmental Panel on Climate Change (IPCC) includes a dedicated task force, separated from the other

three working groups, only for the purpose of greenhouse gas emissions inventory issues (Paustian et al., 2006).

In this instance, there are several emissions inventories that use different input data and approaches for different scales. One of the most frequently used inventories is the Emission Database for Global Atmospheric Research (EDGAR; Olivier



et al. 1996), which provides estimates for the total emissions worldwide. This inventory uses national statistics that do not provide detailed characterizations of high resolution applications. These detailed characterizations are needed for urban studies. There are also continental emissions inventories such as the European Monitoring and Evaluation Programme (EMEP), which compile emissions from the parties of the Convention on Long-range and Transboundary Air Pollution (CLRTAP) (EEA, 2013). Moreover, there is the Regional Emissions inventory in Asia (REAS), which covers China, Japan and other countries (Streets et al., 2003). However, there are many countries and cities that do not include estimates of emissions for environmental and climate planning.

Vehicular emissions are becoming increasingly important in urban centers (Andrade et al., 2017) and measurements have shown that compounds emitted from exhausts can be highly reactive in the atmosphere, contributing to critical episodes of photochemical smog (Nogueira et al., 2015). However, obtaining this type of emissions database can be complicated, since the sources are in movement and the emissions process is complex with many variables. This can be a challenge, especially in developing countries due to the lack of information about the vehicle type, technology, age, motor size, fuel, speeds, accelerations, street type, environmental temperature and humidity, among other aspects. Besides that, there are other aspects involving the emissions inventory. The most common aspects are the accuracy and complexity relating to the exact contribution of the different pollutant sources, and that in most cases, emissions inventories are usually seen as a scapegoat when simulations do not match observation (Pulles and Heslinga, 2010).

Vehicular emissions inventories are classified according to top-down and bottom approaches. Top-down are based on statistics of vehicle composition, representative speeds and country balances, while bottom-up are based on traffic counts, vehicle composition and speed recording (Ntziachristos and Samaras, 2016). The accuracy of the emissions inventory will reflect on the representation of the pollutants in the atmosphere. It is not always related to the complexity of the model. For instance, a meta-analysis of several studies on vehicular emissions (Smit et al., 2010) concluded that there is no evidence that the more complex models perform better than the less complex ones. An emissions inventory must be comprehensive, including all the important sources and aspects regarding the emissions.

All of these complexities were addressed by the Department of Atmospheric Sciences at the University of São Paulo (USP) when modeling the atmospheric chemistry over Brazilian cities using a top-down vehicular emissions inventory with an on-line atmospheric model (Andrade et al., 2015). The Metropolitan Area of São Paulo (MASP), is the most populated megacity in Latin America (IBGE, 2014) and its most important source of pollution comes from the 11 million vehicles that circulate within the Region (CETESB, 2013; DENATRAN, 2015; Andrade et al., 2017). Furthermore, half of all emissions of CO, HC and NOx in the MASP are from vehicles that are more than 10 years old (Andrade et al., 2017). Therefore, we decided to develop our own vehicular emissions inventory model. With this model, we aim to generate scientific estimates and provide useful information to decision-makers and urban/environmental planners.

The main goal of this project was to develop a high spatial and temporal resolution vehicular emissions inventory model, which was named as the VEIN model. The VEIN model follows the bottom-up approach. It allows the classification of vehicles into several categories, different options of emission factors and specification of pollutants, and input traffic from traffic



simulations or other network-based sources. It is capable of producing gridded emissions outputs. It is open source, friendly for the user, and is available to run in any computational platform (Mac, Windows, Linux, etc).

## 2   VEIN model: methodology to estimate vehicular emissions

Temporal and spatial disaggregated emissions are estimated following a general approach of multiplication between activities and emission factors (Pulles and Heslinga, 2010), as shown in Eq. (1).

$$Emission_{pollutant} = \sum_{activity} (AR_{activity} \cdot EF_{pollutant,activity}) \tag{1}$$

where $Emission_{pollutant}$ for any type of pollutant depends on the activity rate $AR$ and the emission factors $EF$, which is the mass of pollutants generated according to the level of activity. In the context of vehicular emissions, $AR_{activity}$ represents the number of vehicles times the distance (km) that they travel. $EF_{pollutant,activity}$ is the emission factor ($g \cdot km^{-1}$) for pollutants of the vehicles.

For a bottom-up estimation of vehicular emissions, a large number of parameters are invoved with the activity (traffic flow, vehicle composition, speed recording, length of road) and emission factors (speed or acceleration dependent including deterioration) (Ntziachristos and Samaras, 2016). In this instance, the following sections provide the theory behind the VEIN model regarding traffic data arrangement, selection of emission factors, emissions estimation, spatial allocation and inputs for atmospheric models.

### 2.1   Traffic data

Traffic data required for the VEIN model must be represented as an hourly amount of vehicles per street.

This traffic data can be provided by traffic simulations, interpolations or by other sources. In the first step, VEIN reads spatial morning rush hour traffic data from each street of a desired area or city. After reading it, VEIN arranges and organizes the data by vehicle composition, according to Eq. (2).

$$F^*_{i,j,k} = Q_i \cdot VC_{i,j} \cdot Age_{j,k} \tag{2}$$

where $F^*_{i,j,k}$ is the vehicular flow at street link $i$ for vehicle type $j$ by age of use $k$. $j$ defines the vehicular composition according to their type of use, type of fuel, size of engine and gross weight, based on definitions of Corvalán et al. (2002). $Q_i$ is the traffic flow at street link $i$. $VC_{i,j}$ is the fraction of vehicles varying according to the type of vehicles $j$ in the composition for street link $i$. $Age_{j,k}$ is the age distribution by vehicular composition $j$ and age of use $k$. This Equation shows that $VC$ splits the total vehicular flow $Q$ to identify the vehicular fraction, which varies according to the type of fuel, size of motor and gross weight. For example, if $Q$ is light duty vehicles (LDV) and we know that 5% of the $Q$ are passenger cars (PC), with engine lesser than 1400 cc, $VC$ is 0.05. This characterization of the fleet depends on the amount and quality of the available information. VEIN then multiplies the traffic with $Age$ to obtain the amount of each type of vehicle by age of use.



Traffic data must be temporally extrapolated because it is usually available for morning rush hour. Traffic data can be estimated from short period traffic count datasets, then expanded to represent longer timespan, such as Annual Average Daily Traffic (AADT; Wang and Kockelman 2009; Lam and Xu 2000). The next step is to extrapolate the vehicular flow at street link $i$, vehicle type $j$, and age of use $k$, to obtain the vehicular flow for hour of the week $l$ ($F_{i,j,k,l}$; see Eq. 3).

$$F_{i,j,k,l} = F_{i,j,k}^* \cdot TF_{j,l} \tag{3}$$

where $TF_{j,l}$ are the temporal factors varying according to each hour of $l$ and type of vehicle $j$. For instance, $TF$ is defined as a matrix with 24 lines and numbers of columns to each day considered, from Monday to Sunday. In order to expand traffic to other hours, $TF$ matrices must be normalized to the hour that represents the traffic data. It means that $TF$ values at morning peak hour must be 1 and the respective proportion must be assigned to the other hours. For example, $TF$ values can be obtained from automatic traffic count stations.

The average speed of traffic flow is very important and it must be determined for each link and hour. Once the vehicular flow is identified for each hour, the average speed is then identified for each hour. This was accomplished by employing curves from the Bureau of Public Roads (BPR; Bureau of Public Roads 1964), as shown in Eq. (4). The process involves calculating speed by dividing the length of road by the time. The time is calculated using the total traffic expanded to each street link $i$ and hour $l$.

$$T_{i,l} = To_i \cdot \left( 1 + \alpha \cdot \left( \frac{Q_{i,l}}{C_i} \right)^{\beta} \right) \tag{4}$$

In Eq. 4, $T_i$ is the travel time per street link $i$ at each hour of the week $l$. $To_i$ is the travel time under free flow conditions where maximum speed was used. $Q_{i,l}$ is the traffic flow at peak hour for each street link $i$ and hour of the week $l$.

$C_i$ is the capacity of vehicles on street link $i$. The parameters $\alpha$ and $\beta$ are adjustments with default values of 0.15 and 4, respectively. These are recommendations by the Bureau of Public Roads (1964). However, the user can use other values. When there is no available information for these calculations such as capacity at each street link $i$, it is possible to apply a simple average between peak and free flow speeds, in order to obtain the average speeds at different hours.

## 2.2 Selection of the emissions factors

The emissions factors describe the relationship intensity of activity and emissions for a given technology (Pulles and Heslinga, 2010). In the case of our model, an emission factor is the mass of pollutant emitted by the vehicular type, technology and years of use. VEIN counts with emission factors for hot and cold exhaust, evaporative, deterioration and wear emissions. VEIN allows three types of hot exhaust emission factors:

1) Speed functions from the Computer programme to calculate emissions from road transport (Copert; Ntziachristos and Samaras 2016), which are stored internally in the model. This approach can be used if there is no local emission factors and if there is information about vehicular speed recordings, simulations, or knowledge of the representative speeds.

2) Emission factors from local sources. The values must be mass per km $g \cdot km^{-1}$ per specific type of vehicle, including fuel type, size and weight, by age of use.





3) Scaled local emission factors with Copert in order to incorporate speed variation for local factors, as shown in Eq. (5). This produces a specific speed dependent on emission factor by age of use for the vehicle.

$$EF_{scaled}(V_{i,l})_{j,k,m} = EF(V_{i,l})_{j,k,m} \cdot \frac{EFlocal_{j,k,m}}{EF(Vdc_{i,l})_{j,k,m}} \tag{5}$$

where $EF_{scaled}(V_{i,l})_{j,k,m}$ is the scaled emission factor and $EF(V_{i,l})_{j,k,m}$ is the Copert emission factor for each street link

$i$, vehicle from composition $k$, hour $l$ and pollutant $m$. $EFlocal_{j,k,m}$ represents the constant emission factor (not speed functions). $EF(Vdc_{i,l})_{j,k,m}$ are Copert emission factors with average speed value of the respective driving cycle for the vehicular category $j$. The São Paulo emission factors data includes recordings of Federal Test Procedure (FTP-75) driving cycle for LDV with an average speed of $34.12 \, \mathrm{km \cdot h^{-1}}$, as shown in the report of driving cycles (Barlow et al., 2009).

By default, VEIN includes deterioration factors from Copert (Ntziachristos and Samaras, 2016). However, it is possible to

include other sources, such as from Corvalán and Vargas (2003).

## 2.3  Emissions estimation

VEIN estimates type of emissions including hot exhaust (EH; Eq. 6), cold start exhaust (EC; Eq. 7), evaporative (EV; Eq. 8), deterioration factors and speciation. The total vehicular emission is the sum of all types of emissions.

### Hot exhaust emission

The VEIN process of emissions estimation is performed per street link, vehicle type, hour of week, and pollutant. Eq. (6) shows the hot exhaust estimation:

$$EH_{i,j,k,l,m} = F_{i,j,k,l} \cdot L_i \cdot EF(V_{i,l})_{j,k,m} \cdot DF_{j,k} \tag{6}$$

In Eq. (6), $EH_{i,j,k,l,m}$ is the emissions for each street link $i$, vehicle category from composition $k$, hour $l$ and pollutant $m$, where $F_{i,j,k,l}$ is the vehicular flow calculated in Eq. 1. $L_i$ is the length of the street link $i$. $EF(V_{i,l})_{j,k,m}$ is the emission factor

of each pollutant $m$. $DF_{j,k}$ is the deterioration factor for vehicle of type $j$ and age of use $k$.

### Cold start emissions

Cold start emissions are produced during engine startup, when the engine and/or catalytic converter system has not reached its normal operational temperature. Several studies have shown the significant impact for these types of emissions (Chen et al.,

2011) (Weilenmann et al., 2009). VEIN also considers cold start emissions - under this condition emissions will be higher, and if the atmospheric temperature decreases, cold start emissions will increase regardless of whether the catalyst has reached its optimum temperature for functioning (Boulter, 1997). For example, studies report that when ambient temperature is -7°C, emissions are one order of magnitude higher than at 22°C (Ludykar et al., 1999).





The VEIN model caters to these emissions by using the approach outlined in Copert (Ntziachristos and Samaras, 2016), as shown in Eq. (7).

$$EC_{i,j,k,l,m} = \beta_j \cdot F_{i,j,k,l} \cdot L_i \cdot EF(V_{i,l})_{j,k,m} \cdot DF_{j,k} \cdot \left( EF_{\text{cold}}(ta_n, V_{i,l})_{j,k,m} - 1 \right) \tag{7}$$

This approach adds two terms to Eq. 6. The first term $EF_{cold}(ta_n, V_{i,l})_{j,k,m} - 1$ is the emission factors for cold start conditions at each street link $i$, vehicle category from composition $k$, hour $l$ and pollutant $m$ and monthly average temperature $n$. Ntziachristos and Samaras (2016) suggest using monthly average temperature. This is an important aspect that will be reviewed in future versions of VEIN.

The second term $\beta_j$ is defined as the fraction of mileage driven with a cold engine/catalyst (Ntziachristos and Samaras, 2016). The VEIN model incorporates a dataset of cold starts recorded during the implementation of the International Vehicle Emissions (IVE) model (Davis et al., 2005) in São Paulo (Lents et al., 2004), which provides the hourly mileage driven with cold start conditions.

**Evaporative emissions**

Evaporative emissions are important sources of hydrocarbons and these emissions are produced by vaporization of fuel due to variations in ambient temperatures (Mellios and Ntziachristos, 2016; Andrade et al., 2017). There are mainly three types of evaporative emissions: diurnal emissions, due to increases in atmospheric temperature, which lead to thermal expansion of vapor fuel inside the tank; running losses, when the fuel evaporates inside the tank due to normal operation of the vehicle; and hot soak emissions, which occur when the hot engine is turned off. These methods implemented in VEIN were sourced from the evaporative emissions methods of Copert (Mellios and Ntziachristos, 2016). This approach is shown in Eqs. (8), (9) and (10).

$$EV_{j,k} = \sum_s D_s \cdot \sum_j F_j \cdot (HS_{j,k} + de_{j,k} + RL_{j,k}) \tag{8}$$

where $EV_j$ are the volatile organic compounds (VOC) evaporative emissions due to each type of vehicle $j$. $D_s$ is the "seasonal days" or number of days when the mean monthly temperature is within a determined range: [-5°,10°C], [0°, 15°C], [10°, 25°C] and [20°, 35°C]. $F_{j,k}$ is the number of vehicles according to the same type $j$ and age of use $k$. $HS_{j,k}$, $de_{j,k}$ and $RL_{j,k}$ are average hot/warm soak, diurnal and running losses evaporative emissions $(g \cdot day^{-1})$, respectively, according to the vehicle type $j$ and age of use $k$. $HS_{j,k}$ and $RL_{j,k}$ are obtained using equations also sourced from Mellios and Ntziachristos (2016):

$$HS_{j,k} = x_{j,k} \cdot (c \cdot (p \cdot e_{shc} + (1-p) \cdot e_{swc}) + (1-c) \cdot e_{shfi}) \tag{9}$$

where $x$ are the number of trips per day for the vehicular type $j$ and age of use $k$. $c$ is the fraction of vehicles with fuel return systems. $p$ is the fraction of trips finished with hot engine, for example, an engine that has reached its normal operating temperature and the catalyst has reached its light-off temperature (Ntziachristos and Samaras, 2016). The light-off temperature is the temperature at the point when catalytic reactions occur inside a catalytic converter. $e_{shc}$ and $e_{swc}$ are average hot-soak and





warm-soak emission factors for gasoline vehicles with carburettor or fuel return systems $(\mathrm{g} \cdot \mathrm{parking}^{-1})$. $e_{shfi}$ is the average hot-soak emission factors for gasoline vehicles equipped with fuel injection and non-return fuel systems $(\mathrm{g} \cdot \mathrm{parking}^{-1})$.

$$RL_{j,k} = x_{j,k} \cdot (c \cdot (p \cdot e_{rhc} + (1-p) \cdot e_{rwc}) + (1-c) \cdot e_{rhfi}) \tag{10}$$

$x$ and $p$ have the same meanings of Eq. 9. $e_{rhc}$ and $e_{rwc}$ are average hot and warm running losses emission factors for gasoline
vehicles with carburettor or fuel return systems $(\mathrm{g} \cdot \mathrm{trip}^{-1})$ and $e_{rhfi}$ are average hot running losses emission factors for gasoline vehicles equipped with fuel injection and non-return fuel systems $(\mathrm{g} \cdot \mathrm{trip}^{-1})$. It is recommended to estimate the number of trips per day (Mellios and Ntziachristos, 2016), $x$, as the division between the mileage and 365 times the length of trip: $x = \dfrac{mileage_j}{(365 \cdot ltrip)}$. However, the mileage of a vehicle is not constant throughout the years. Therefore, VEIN incorporates a dataset of equations to estimate mileage of different types of vehicles by age of use (Bruni and Bales, 2013).

**2.4   Speciation of emissions in chemical sub-components**

Particulate matter and hydrocarbons are a mixture of several chemical compounds that play an important role in atmospheric chemistry (Seinfeld and Pandis, 2016). VEIN includes speciation profiles for hydrocarbons and particulate matter from Ntziachristos and Samaras (2016) and Rafee (2015). These profiles are percentages of the emissions by vehicle type, fuel, emission standard and other characteristics. Also included are speciations of particulate matter in black carbon and organic matter,
particulate matter fractions for tire, brake and road wear, non-methanic hydrocarbons and nitrogen oxides.

**2.5   Spatial allocation and data-bases**

VEIN provides functions to generate grids and spatially allocate emissions into grids. This is helpful for visualization and generation of inputs for atmospheric models, and as a tool for urban planning. In addition, VEIN includes functions to produce a database of hourly emissions for vehicular composition by age of use. Section 4.4 provides details and examples about the
emissions grids and databases.

**3   VEIN model design**

The VEIN model was constructed using the free open source R software (R Core Team, 2017). R is a programming language and environment for statistical computing and graphics (R Core Team, 2017). It was developed primarily for analyzing data. However, since its capabilities have grown over time, R has become a flexible language with many different areas of application.
It includes elements of programming language such as Lisp and syntax of S, as described by Leiner et al. (1997).

The VEIN R package depends on the package **sp** (Bivand et al., 2013), as it uses several of its classes. VEIN imports some functions from the package **rgeos** (Bivand and Rundel, 2016), which is an interface for the Geometry Open Source (GEOS) library (https://trac.osgeo.org/geos/). It also imports functions from **rgdal** (Bivand et al., 2016), which provides bindings to the Geospatial Data Abstraction Library (GDAL; http://www.gdal.org/). Therefore, these R packages must be installed prior to
using the VEIN package.

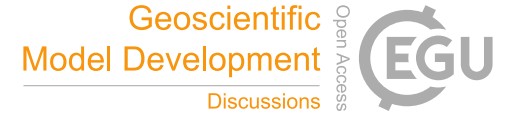

VEIN started between 2014 and 2016 as a collection of several R scripts, initially named R-EMIssions (REMI; Ibarra-Espinosa and Ynoue 2017), which later evolved into an R package. It was developed in R due to the free open source advantages and because R allows easier reproducibility. VEIN is open to scrutiny from its community of users, thus allowing opportunities for user feedback and improvements. This facilitates widespread use of the model and identifying any software bugs/errors,

with potential for adding new capabilities. VEIN has its own functions, but it also incorporates other data and functions such as emission factors and mileage.

VEIN can be installed from the Comprehensive R Archive Network (CRAN) https://CRAN.R-project.org/package=vein or from the website Github https://github.com/ibarraespinosa/vein. In order to use the VEIN library and run the demo, it is necessary to run the following scripts in R:

```
install.packages("vein") #or
        library(devtools)
        install_github("ibarraespinosa/vein")

        library(vein)
demo(VEIN)
```

The diagram process for estimating emissions is shown in Fig. 1. The green circles in this Figure refer to the data and the blue boxes refer to the functions inside the model. The VEIN model diagram starts at the green circle traffic, which represents the morning rush hour traffic data for each street link. Then the *age* functions (*age_ldv*, *age_hdv*, *age_ldv* or *my_ldv*) determine the

vehicular composition by age of use as shown in Eq. (2). The data *profile* allows to temporally extrapolate traffic data to the other hours and this allows to estimate the average vehicular speed to any hour and link using the function *netspeed*. Emission factor selections start by adding the deterioration effect with the function *emis_det* into local, speed dependent emission factors from Ntziachristos and Samaras (2016), denoted as $speed\_ef$ or scaled emission factors denoted as $scaled\_ef$ in Fig. 1. Besides including speed dependent emission factors from Ntziachristos and Samaras (2016), VEIN also includes local emission factors

from CETESB (2016). Once the input data is ready, the function *emis* estimates hourly emissions for each hour of the day, and day of the week. The function *emis_post* produces an emissions database by vehicle category or by street, denoted as df and street in Fig. 1 respectively. These emissions are then speciated with the function *speciate*. At this time, the user can create a grid with the function *make_grid*, which creates a rectangular grid with format `SpatialPolygonsDataFrame` or a `SpatialGridDataFrame` used to allocate emissions spatially with the function *emis_grid*. The function *emis_wrf* reads

the emissions grids and creates a data-frame ready to create an input for WRF-Chem model (Grell et al., 2005). This data-frame must be exported as a .txt file and could be used as an input into other atmospheric models such as BRAMS (Freitas et al., 2005).





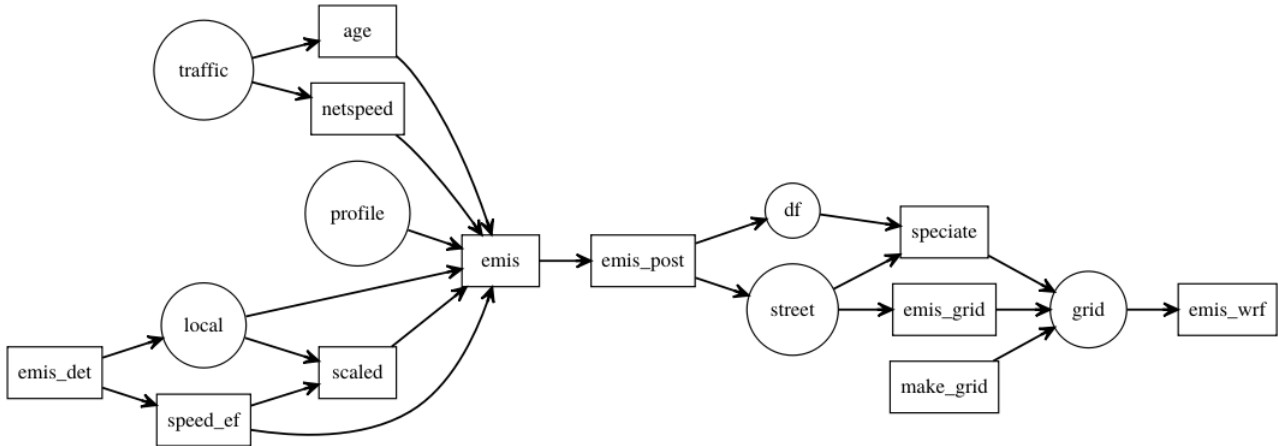

**Figure 1.** Representation of the VEIN model. Boxes and circles represent functions and data, respectively.

### 3.1 Functions and classes

VEIN uses objects of class **Spatial** (Pebesma and Bivand, 2005), including `SpatialLinesDataFrame`, to represent road segments. To read geospatial data, there are several packages,

such as **rgdal** (Bivand et al., 2016) or **maptools** (Bivand and Lewin-Koh, 2015). The main requirement is that the network
must be a `SpatialLinesDataFrame`, class of **sp** (Pebesma and Bivand, 2005).

We included several functions to arrange traffic data, select or scale emission factors, as well as estimate and process emissions in VEIN, as shown in Table 1. These functions implement the equations shown in Section 2.

VEIN incorporates 8 classes (see Table 1), which are objects with specific characteristics: methods and units. The methods are print, summary and plot. They are functions that return a specific result depending on each class. Another important
characteristic of each class is that they include explicit units, in an effort to reduce human errors and improve the usability. For this task, VEIN imports some functions of the package **units** (Pebesma et al., 2016), which is an interface in the C library **udunits** from University Corporation for Atmospheric Research (UCAR). Therefore, this library must be installed on the system prior to using VEIN. Only the `EmissionFactorsList` and `EmissionsArray` do not show their units explicitly due to limitations with the units package. The classes outlined in Table 1 are also constructor functions, which means that
they can create VEIN classes and add the respective units. VEIN incorporate constructor functions to create classes such as `Vehicles` or `Emissions`. These functions are incorporated inside other VEIN functions in order that the output of VEIN has a class. When the constructor function are applied to a numeric element, the constructor simply adds the units and the resulting object has class **units**. For example, applying the function `EmissionsArray` to a numeric vector will add the units $g \cdot h^{-1}$ to the numeric vector.





Table 1: Summary of the VEIN classes, functions and internal data.

| Function | Description | Reference |
|---|---|---|
| age_hdv | Distribution of HDV by age of use | Ministério do Meio Ambiente (2011) |
| age_ldv | Distribution of LDV by age of use | Ministério do Meio Ambiente (2011) |
| age_moto | Distribution of Motorcycle by age of use | Ministério do Meio Ambiente (2011) |
| ef_evap | Evaporative emission factors | Mellios and Ntziachristos (2016) |
| ef_hdv_scaled | List of scaled emission factors for HDV | Ntziachristos and Samaras (2016) |
| ef_hdv_speed | HDV Emission factors | Ntziachristos and Samaras (2016) |
| ef_ldv_cold | LDV cold start emission factors | Ntziachristos and Samaras (2016) |
| ef_ldv_cold_list | List of LDV cold start emission factors | Ntziachristos and Samaras (2016) |
| ef_ldv_scaled | List of scaled emission factors for LDV | Ntziachristos and Samaras (2016) |
| ef_ldv_speed | LDV Emission factors | Ntziachristos and Samaras (2016) |
| ef_wear | Tyre and break wear, and road abrassion | Ntziachristos and Boulter (2009) |
| EmissionFactors | Creates class `EmissionFactors` ($g \cdot km^{-1}$) | |
| EmissionFactorsList | Creates class `EmissionFactorsList` ($g \cdot km^{-1}$) | |
| Emissions | Creates class `Emissions` ($g \cdot h^{-1}$) | |
| EmissionsArray | Creates class `EmissionsArray` ($g \cdot h^{-1}$) | |
| EmissionsList | Creates class `EmissionsList` ($g \cdot h^{-1}$) | |
| emis | Estimation to hour and day of the week | |
| emis_cold | Cold start estimation | Ntziachristos and Samaras (2016) |
| emis_det | Deterioration factors | Ntziachristos and Samaras (2016) |
| emis_evap | Evaporative estimation | Mellios and Ntziachristos (2016) |
| emis_grid | Allocation on rectangular grid | |
| emis_paved | Resuspenssion of paved roads | USA-EPA (2016) |
| emis_post | Post processing of emissions | |
| emis_wear | Estimation of wear emissions | Ntziachristos and Boulter (2009) |
| emis_wrf | Creating data-frame to WRF-Chem | Vara-Vela et al. |
| Evaporative | Creates class `Evaporative` ($g \cdot d^{-1}$) | |
| fe2015 | Data of CETESB emission factors | CETESB (2016) |
| fkm | Data of mileage functions by vehicle | Bruni and Bales (2013) |
| hot_soak | Hot soak evaporative | Mellios and Ntziachristos (2016) |
| make_grid | Rectangular grid | |
| my_age | Distribution of vehicles by age of use | |



Table 1: Summary of the VEIN classes, functions and internal data.

| Function | Description | Reference |
|---|---|---|
| net | Data of traffic simulation of west São Paulo | CET (2014) |
| netspeed | Estimate average speed | |
| pc_profile | Data of temporal factors | ARTESP (2012) |
| pc_cold | Data of vehicle start pattern | Lents et al. (2004) |
| running_losses | Evaporative estimation | Mellios and Ntziachristos (2016) |
| speciate | Split by species | Ntziachristos and Samaras (2016), Rafee (2015) |
| Speed | Creates class Speed $(km \cdot h^{-1})$ | |
| temp_fact | Expand hourly traffic | |
| Vehicles | Creates class Vehicles $(1 \cdot h^{-1})$ | |
| vkm | Determination of vehicle-kilometers | |

## 4 Estimating MASP vehicular emissions using VEIN model

This section presents the application of the most important functions of the VEIN model. These functions obtain an estimate of CO emissions from LDV fleets in MASP for 2015 (for a typical non-holiday week).

### 4.1 Traffic data for MASP

Hourly traffic is a requirement for this data. This data can only be represented as one hour of data, which can then be extrapolated with the VEIN functions or as a list of hourly traffic data for all the covered hours. The present application includes a morning rush hour traffic simulation for MASP CET (2014) loaded into R as a SpatialLinesDataFrame. It includes peak and free flow speeds, along with capacity (maximum amount of vehicles that can circulate in a road per hour) and traffic flow from LDV and HDV. Fig. (2) shows the traffic simulation of LDV at 08:00-09:00 local time (LT), where urban motorways

concentrate the highest amount of vehicles. The total volume of LDV is 24708767 $veh \cdot h^{-1}$ and the number of streets are 34733 with mean of 711 $veh \cdot h^{-1} \cdot street^{-1}$. It is important to note that the VEIN model provides an extraction of the traffic simulation for the western part of São Paulo. The traffic simulation for MASP weights 61.6 Mb and the extraction for the western part of São Paulo 3.1 Mb. We provided the extraction and not the whole traffic simulation in VEIN to make it faster. This section provides codes to run VEIN so that the reader can follow them with the data provided in the model.

After loading traffic data, the traffic flow was expanded to each hour of the week with the function *temp_fact*, as shown in the following scripts. It is also necessary to extrapolate hourly vehicle speeds. Therefore, we created the function *netspeed*, which applies the function BPR (Bureau of Public Roads, 1964) curves, according to the Eq. 4. To use BPR, a data-frame is required





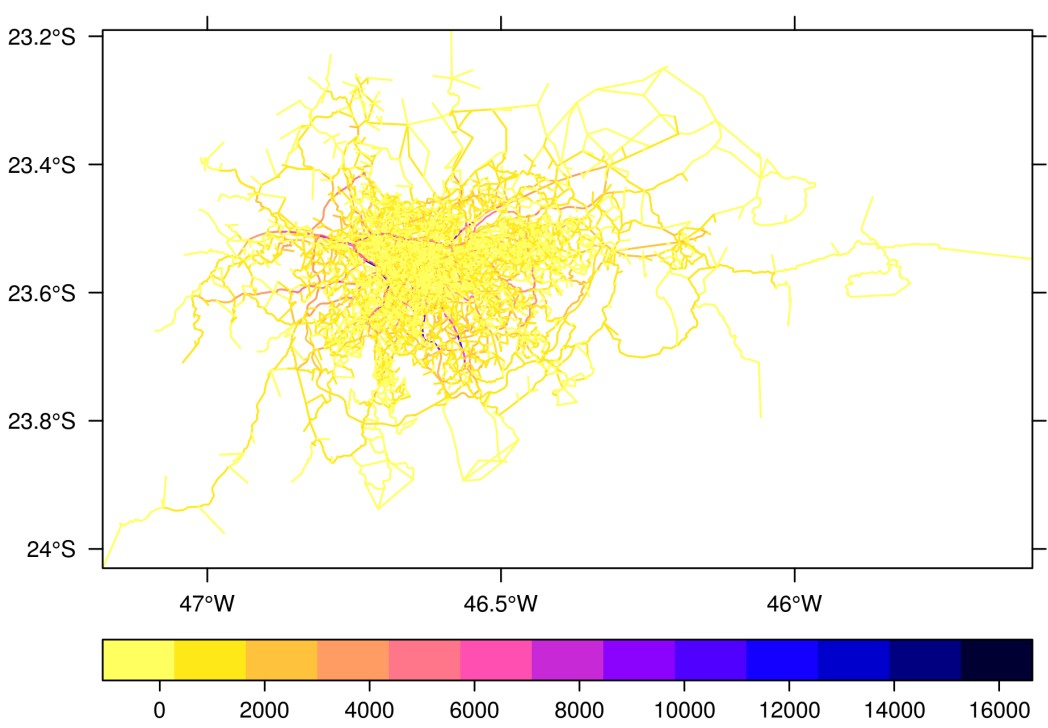

**Figure 2.** Traffic flow simulation for LDV $(\mathrm{veh}\cdot\mathrm{h}^{-1})$ at 08:00-09:00 LT for MASP.

with total traffic at all hours and the morning rush parameters capacity, peak speed, free flow speed, length of the road, and with BPR parameters alpha and beta. The argument *scheme* produces a 24 hour speed data-frame, based only on peak and free flow speed with a profile of free flow speeds at early mornings, peak speeds, morning and evening rush hours, and the average at hours in between. If the time-lapse for the emissions estimation is longer than a week, the user could simply replicate the hours until it reaches the desired hours.

```
data(net)
data(pc_profile)
pcw <- temp_fact(net$ldv+net$hdv, pc_profile)
speedspeed <- netspeed(pcw, net$ps, net$ffs, net$capacity, net$lkm, alpha = 1)
```

For example purposes, the resulting speeds can be observed in Fig. 3, which shows two different speed maps: one for 08:00 LT (left panel) and the other for 23:00 LT (right panel). This Figure shows that the highest speeds are found in most of the streets further away from the MASP center at both times (08:00 and 23:00 LT). The major difference between the two panels





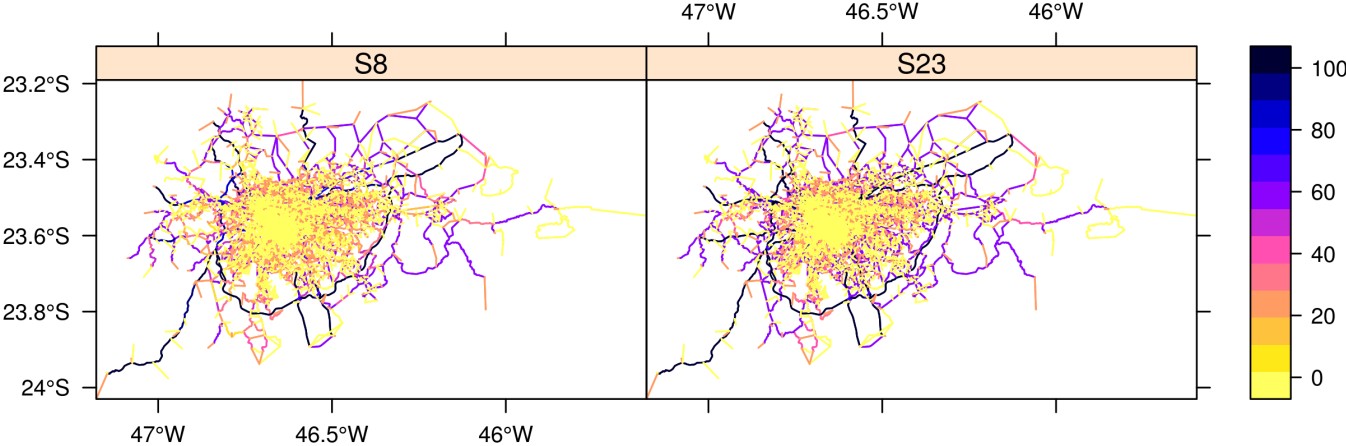

**Figure 3.** Traffic speeds (colored lines; $km \cdot h^{-1}$) for LDV fleet at 08:00 LT (left panel) and 23:00 LT (right panel) in MASP.

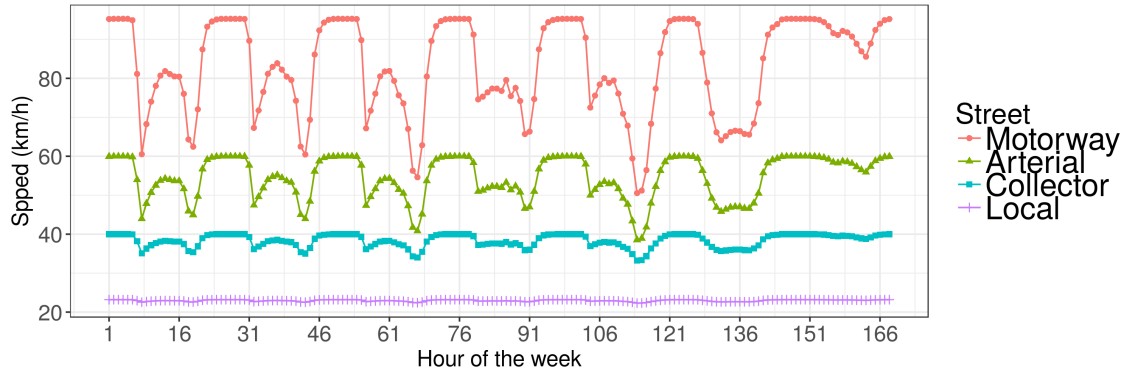

**Figure 4.** Traffic average speeds for LDV fleet by type of street (colored lines) at 08:00-09:00 LT in MASP.

(left and right), is that in the late night, the flow is faster near the center of MASP. This seems reasonable since the vehicular flow tends to diminish during the night. The average speeds also show a pattern related to the type of street as shows Fig. 4. The type of streets comes form São Paulo traffic simulation and they were translated to English and are defend as Motorway: roads with speed limits above 80 $km \cdot h^{-1}$ without physical intersections. Arterial: roads with speed limit of 60 $km \cdot h^{-1}$ with

5    intersection such as traffic lights. Collector: roads with speed limit of 40 $km \cdot h^{-1}$ that collect and distribute traffic between Arterial streets. Local: roads with speed limit of 30 $km \cdot h^{-1}$ that access restricted zones. Figure 4 shows that lower speeds are found during the morning (07:00-10:00 LT) and evening (17:00-20:00 LT) rush hours. This is important in terms of air pollution because at lower speeds vehicles emit more pollutants (Ntziachristos and Samaras, 2016). On the contrary, maximum average speeds for each type of road are obtained at night hours and on Sundays at all hours.



After calculating the São Paulo traffic flow average speeds for each hour of the week and each street link, the age distribution of the fleet was obtained by type of vehicle. The *age\** functions (*age_ldv*, *age_hdv* and *age_moto*) distribute the traffic data by the vehicle's age of use. These functions return a data-frame with the number of rows matching the number of streets, columns representing the amount of vehicles by age of use, and a message indicating the average age of the fleet. The *age\** functions

are related to the Eq. 2, where they split the vehicular flow at street link $Q_i$ by type of vehicle $j$ and the vehicle's age of use $k$. These functions are based on the survival equations presented in the Brazilian Emissions Inventory Report by the Ministério do Meio Ambiente (2011) and parameterized for the VEIN model. They allow the use of different coefficients to obtain different age distributions allowing the representation of different realities. Furthermore, the function *my_age* distributes the traffic from an existing dataset, e.g. yearly vehicle licensing.

The following code shows three uses of *age\** functions. The first, *my_age*, uses yearly traffic data from the report of the São Paulo emissions inventory (CETESB, 2016), expressed as CETESB_PC based on vehicle sales. The second, *age_ldv*, uses default parameters, and the third, *age_ldv*, uses b = -0.14.

```
CETESB_PC <- c(33491,22340,24818,31808,46458,28574,24856,28972,37818,49050,87923,
133833,138441,142682,171029,151048,115228,98664,126444,101027, 84771,55864,36306,
21079,20138,17439, 7854,2215,656,1262,476,512, 1181, 4991, 3711, 5653, 7039,5839,
4257,3824,3068)
pc1 <- my_age(x = net$ldv, y = CETESB_PC, name = "PC")
pc2 <- age_ldv(x = net$ldv, name = "PC", agemax = 41)
pc3 <- age_ldv(x = net$ldv, name = "PC", b = - 0.14, agemax = 41)
```

Figure 5 (a) shows three age distributions, each one sums 24708767 $\text{veh} \cdot \text{h}^{-1}$, and each have different average age. It represents São Paulo LDV fleet according to its age of use, with an estimated average age of 11.09 years (red line), 15.53 years (blue line), and 15.17 years (green line ). This Figure shows that in MASP there are more newer vehicles than older ones. *age\** functions also include a logical option named *bystreet*, with default value equal to FALSE. When this value is TRUE, *age\** expects that the coefficients *a* and *b* for the *age\** functions are numeric vectors, with length matching the number of streets.

This allows different age distributions within the same road network and it is particularly useful for areas with less information about the vehicles' age of use.

## 4.2    Emission Factors

Once we obtain the traffic flow for the desired type of vehicles (in our LDV example), for each hour of the day, for all (desired) days of the week, for each age distribution and for each street link, then we can proceed to the emissions calculation by itself.

The VEIN package includes a database titled *fe2015* with emission factors for PC and light trucks by age of use from the São Paulo official vehicular emissions inventory (CETESB, 2016). This inventory was performed using a top-down approach and the pollutants estimated were $CH_4$, CO, $CO_2$, HC, $N_2O$, NMHC, $NO_X$, and PM. This data includes national and equivalent Euro Emission Standards by year and age. The equivalence among Brazilian CONAMA (1986), MMA (2011), MMA (2015)



**Table 2.** Proposed equivalence of emission standards used in São Paulo study.

| Vehicle | Brazilian Standard | Euro Standard | Year |
|---|---|---|---|
| | L - 1 | Pre ECE | 1988 - 1991 |
| | L - 2 | Euro 1 | 1992 - 1996 |
| | L - 3 | Euro 2 | 1997 - 2004 |
| LDV | L - 4 | Euro 3 | 2005 - 2011 |
| | L - 5 | Euro 4 | 2012 - 2013 |
| | L - 6 | Euro 5 | 2014 |
| | P - 1 | Pre Euro | 1990 - 1992 |
| | P - 2 | Pre Euro | 1993 |
| | P - 3 | Euro 1 | 1994 - 1997 |
| HDV | P - 4 | Euro 2 | 1998 - 2003 |
| | P - 5 | Euro 3 | 2004 - 2011 |
| | P - 6 | Euro 4 | - |
| | P - 7 | Euro 5 | 2012 |
| | M - 1 | Euro 1 | 2003 - 2005 |
| Motorcycle | M - 2 | Euro 2 | 2006 - 2008 |
| | M - 3 | Euro 3 | 2009 |

Based on http://transportpolicy.net/index.php?title=Category:Brazil

and Euro Directive70/220/EEC (1991) was added into this database in order to choose the corresponding matching vehicle and emissions standard. The equivalence can be seen in Table 2.

*fe2015* emission factors do not include the deterioration effect due to accumulated age of vehicle and it must be included. This is performed with the the deterioration factor function *emis_det* which has the arguments: pollutant, size of engine, Euro Standard and mileage in km. VEIN includes a Brazilian database of mileage functions named *fkm*, which is a list of functions with each element of the list corresponding to vehicle type. These functions depend on the vehicle's age of use and they originate from the odometer readings of more than $1.6 \cdot 10^6$ vehicles (Bruni and Bales, 2013).

Emission factors for PC, LCV and motorcycles are called with the function *ef_ldv_speed*. In the case of trucks and buses they use the function *ef_hdv_speed*. The arguments are filters for an internal database of emission factors which include several parameters such as fuel, Euro Standard, volume of engine and load, among others. These functions also include a multiplication factor with a default value of 1. Exact spelling is required when using the arguments. If the argument names are entered incorrectly, VEIN will not return the emission factor functions.

The following code shows how to read the emission factors of the VEIN databases $fe2015$, $pc\_profile$ and $fkm$, in order to incorporate the deterioration effect into the CETESB (2016) emission factors. The age of LDV shown in Fig. 5 (a) has a length of 41 years. This means that it needs 41 emission factors, one per each age of use. It calls the function *emis_det*, which requires the accumulated mileage, obtained from the list of mileage equations $fkm$. The Fig. 5(c) shows the emission factors



from CETESB with and without deterioration by age of use. We are using deterioration factors from Ntziachristos and Samaras (2016) that affect only vehicles with a catalytic system. The base year of this emissions estimation is 2015 and the vehicles with catalytic system started in 1992 in Brazil (23 years before 2015). Therefore, the vehicles that entered into the market before 1992 do not include deterioration. The emission factors dataset $fe2015$ includes emission factors for vehicles with only

36 years of use but the vehicular distribution calculated in the last script has 41 years of use. Therefore, we repeated the oldest emission factors to have 41 emission factors. Here we are assuming that the emission factors of vehicles 36 years of use it is the same as the vehicles till 41 years of use. The last line of the following script calculates the deteriorated emission factors of Passenger Cars by age of use.

```
     data(fe2015)
data(pc_profile)
     data(fkm)
     pckm <- fkm[[1]](1:24)
     pckma <- cumsum(pckm)
     cod1 <- emis_det(po = "CO", cc = 1000, eu = "III", km = pckma[1:11])
cod2 <- emis_det(po = "CO", cc = 1000, eu = "I", km = pckma[12:24])
     co1 <- fe2015[fe2015$Pollutant == "CO", ]
     co1[37:41, ] <- co1[36, ]
     cod <- c(co1$PC_G[1:24] * c(cod1, cod2), co1$PC_G[25:nrow(co1)])
```

Once the deterioration effect was added into the Brazilian emission factors CETESB (2016), they were scaled to account for
speed with the function *ef_ldv_scaled*. This function is used to multiply emission factors from *ef_ldv_speed* with a constant.

The new emission factor (dependent on speed) has the same value as the local emission factor, which is evaluated at the reference speed of the measurement $34.12 \, \mathrm{km \cdot h^{-1}}$ for FTP-75. The default speed value is $34.12 \, \mathrm{km \cdot h^{-1}}$, but this value must change correspondingly to the speed of the driving conditions. To use this function, it is necessary to scale emission standards of local emission factors with Euro Standards. In the following code, *Euro_LDV* is a vector indicating Euro Standard by age of
use.

```
     lef <- ef_ldv_scaled(co1, cod, v = "PC", cc = "<=1400", f = "G", p = "CO",
     eu = co1$Euro_LDV)
```

### 4.3 Emission estimation

After inputting the database of vehicles and their respective emission factors, VEIN is ready to use the *emis* function. The VEIN
package counts with several *emis* functions according to the type of emission being estimated. The *emis* function assembles data and outputs from other VEIN functions, and estimates the emissions for the number of hours and days in the week. This function reads the morning rush hour traffic data by age of vehicle use and extrapolates it with the profile data-frame, as previously explained. It reads the emission factors stored in a list with length matching the age distribution of the vehicle





category and then reads the list of speeds. This function returns the emissions at each street in an array with 4 dimensions: 1) number of streets; 2) max age of age distribution; 3) hours (usually 24); and 4) days (usually 7). For convenience, there are defined default values for this function hour = 24, day = 7 and array = TRUE. The values can be changed accordingly.

For example, the estimation of the traffic simulations shown in Fig. 2 has 34,733 streets, fleet of 41 years age distribution, 24 hours of the day, and 7 days of the week. Therefore, it will produce an emissions array with dimensions 34,733, 41, 24 and 7. The vehicle fleet used to produce the age distribution is shown in green in Fig. 5(a) and it has 41 years of length.

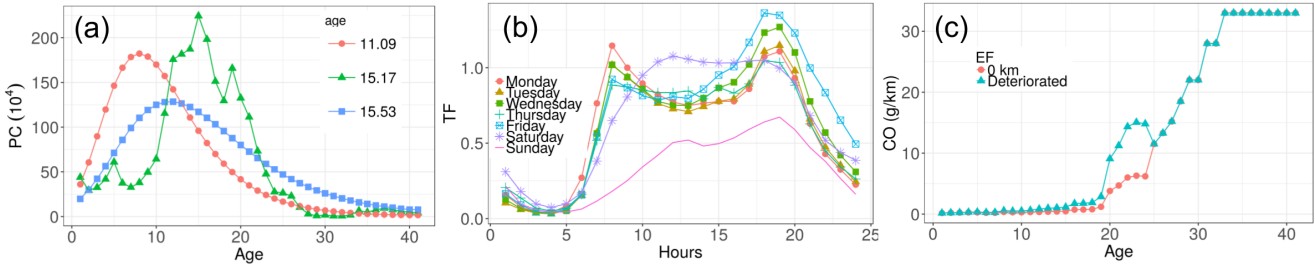

**Figure 5.** (a) Distribution of LDV composition by age of use, (b) temporal factor for expanding morning rush hour traffic data and (c) CO emission factors used in the estimation presented in this manuscript.

```
data(pc_profile)
E_CO <- emis(veh = pc1, lkm = net$lkm, ef = lef, speed = speed, profile = pc_profile)
```

This emissions array output for 34,733 streets and vehicle fleet with 41 years age distribution, 24 hours and 7 days of the week, has the size of 1.8 Gb. Hence, it is recommended to use the function *emis_post* and then delete the original emissions array. The arguments include: the emissions array, type of vehicle, size or weight, fuel, pollutant, and the boolean argument *by*. The *emis_post* function was created to preserve the most important information in the emissions array, to use less memory size and to be compatible with the packages **sp** (Pebesma and Bivand, 2005) and **ggplot2** (Wickham, 2009). VEIN outputs could also be

used with the package **openair** (Carslaw and Ropkins, 2012). *emis_post* returns a data-frame, but the argument *by* determines the shape of the data-frame. When *by* has the value 'veh', it returns a data-frame with an aggregation of the emissions array by each vehicle's age of use with columns: vehicle name, emission (in grams), vehicle type, size, fuel, pollutant, age, hour and day. This output allows the user to visualize hourly emissions at each day of the week, as shown in Fig. 6. Higher emissions are found at morning and evening rush hours from Monday to Friday. Saturday has peak higher emissions at noon and Sunday

has the lowest emissions.

VEIN enables the user to identify which type of vehicle emits more by age of use. This is particularly useful for environmental authorities who aim to reduce local traffic emissions and restrict the circulation of high-emitting vehicles. Fig. 6 shows the CO emissions of gasoline fueled LDV by the vehicle's age of use. The average age of these vehicles is 15.17 years, as shown by the green curve in Fig. 5 (a). The total number of vehicles is 24708767 $\text{veh} \cdot \text{h}^{-1}$ (8:00-9:00 LT on a Monday). The

total CO emissions is 233095 $\text{t} \cdot \text{y}^{-1}$, considering a year of 52 weeks, but the emissions are concentrated for the LDV between



20 and 23 years of use. The vehicles in this age interval represent 14.76% of the fleet, emitting 63.79% (148712 $t \cdot y^{-1}$) of the total emissions. In other words, 15% of the fleet emit more than 60% of the CO. Between 1992 and 1996, the emissions standard was Proconve L2, equivalence with Euro 1 (see Table 2), and also the introduction of the catalytic system in Brazil. Therefore, the high emissions are due to vehicles with a deteriorated catalytic converter. This information is useful for reducing

5   air pollution, thus supporting the aims of environmental planners and local authorities.

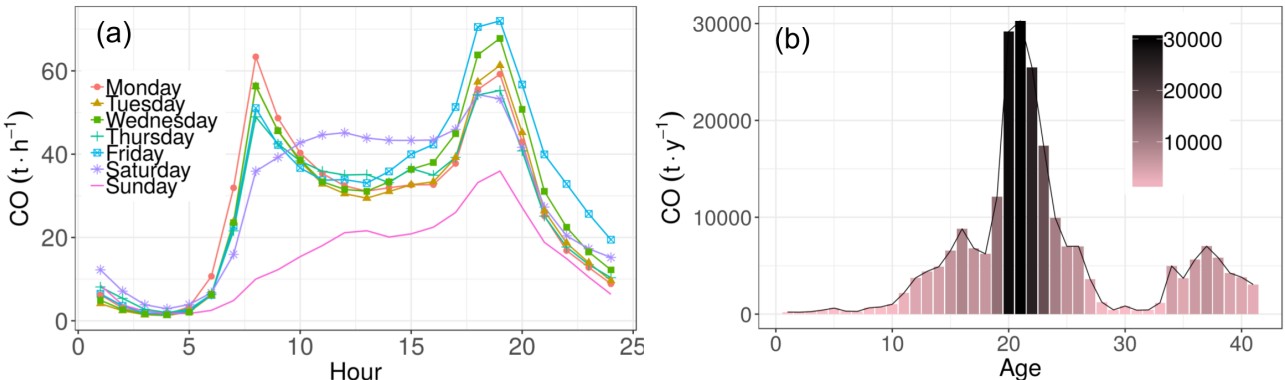

**Figure 6.** (a) CO Emissions $(g \cdot h^{-1})$ per hour of the day and day of the week (colored and shaped lines) for LDV from MASP, (b) CO Emissions $(t \cdot y^{-1})$ according to the age of use of the LDV from MASP.

### 4.4 Post estimation

The spatial dimensions of the emissions estimation is an important feature of VEIN because it allows the representation of the streets into spatial vectors. This is accomplished by using the function *emis_post* with the argument *by* equal to 'streets_narrow' or 'streets_wide'. Both options return a data-frame with different characteristics, which can be converted into spatial vectors.

10   When *by* is equal to 'streets_narrow', it returns a data-frame with four columns: id, indicating the number of rows, emissions, hour and day of the week. The number of rows in the data-frame is the original number of selected streets multiplied by the hours and days of the week. For example, when there are 34733 streets, 24 hours and 7 days of the week, it returns a data-frame with 5835144 rows having a size of 133.6 Mb. This option is useful to visualize the temporal behavior of specific streets with **ggplot2** (Wickham, 2009) or **ggmap** (Kahle and Wickham, 2013), for instance.

15     In most cases, users will be particularly interested when the argument *by* is equal to 'streets_wide'. This produces a data-frame with number of rows matching number of streets for the domain and number of columns as hours. Fig. 7 shows the CO emissions for LDV at each street on a Friday at 19:00 LT. The following code shows how to produce hourly emissions by street and then add these emissions back into the `SpatialLinesDataFrame` net. This is possible because the number of rows in `E_CO_STREETS` is equal and it matches the number of rows in `net`.

```
E_CO_STREETS <- emis_post(arra = E_CO, pollutant = "CO", by = "streets_wide")
net@data <- cbind(net@data, E_CO_STREETS)
```





The emissions shown in Fig. 7 are concentrated in two streets, a motorway and a trunk street at the northern part of the emissions map. This image was generated with the function **spplot** in the package **sp**.

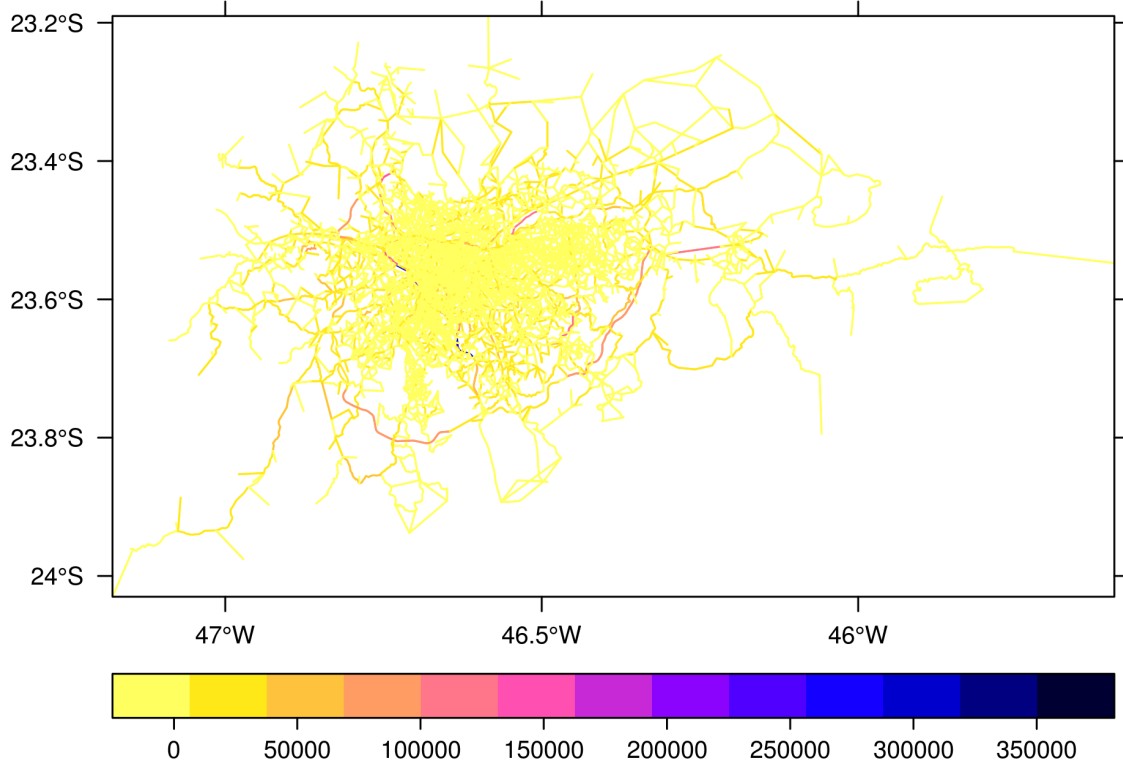

**Figure 7.** CO Emissions (colored lines; $g \cdot h^{-1}$) for LDV at Friday 19:00 LT over MASP.

It also shows a rectangular grid, which can be used for allocating the emissions. The allocation of emissions into the grid is very important for visualization and for inputs to air quality models. We included a simple function to create a rectangular
5  grid in VEIN. The function was named, *make_grid*, which has the arguments, *width*, *height* and a boolean argument *polygon* for determining the type of output. When the argument *polygon* is TRUE, it returns a `SpatialPolygonsDataFrame`, and when it is FALSE, it returns a `SpatialGridDataFrame`. The units of 'width' and 'height' depend on the coordinate reference systems of the data.

The allocation of emissions in each grid cell is produced by a spatial interception between the emissions at each street and
10  the polygon-grid. Firstly, the `SpatialLinesDataFrame` object with emission must contain a column with the length of the street. The length is calculated with the function *gLength* in the package **rgeos** (Bivand and Rundel, 2016). Secondly, it is performed at the intersection between the `SpatialLinesDataFrame` of emissions and the grid `SpatialPolygonsDataFrame`. The intersection is performed by importing the function *intersect* in the package **raster** (Hijmans, 2016). The grid must have a column with the id for each cell. Thirdly, it calculates, in another column, the length of the street in the resulting

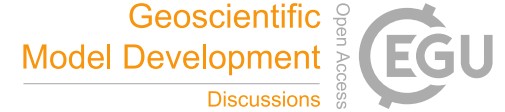

`SpatialPolygonsDataFrame`. Then it multiplies the emissions with the proportion of the new and old length of the street. This allows proportional emissions in each grid cell. Fourthly, it aggregates the emissions by id of grid and adds these emissions by grid id into the grid. The results are in emissions grid with format `SpatialPolygonsDataFrame`. These calculations can be performed automatically by the function *emis_grid*.

The function *make_grid* is suitable in mid-size or small cities when the resolution is approximately 1 km. When dealing with larger cities and higher resolution, it is recommended to use other tools because *make_grid* would take up too much time. This difficulty will be overcome in a future version of VEIN with dependencies on package **sf** (Pebesma, 2016). In the following code, we show the use of the function *make_grid* only for example purposes. It is recommended that function *emis_post* be used with the argument 'by = streets_wide', in order to return a data-frame with hourly emissions for each street. This output

can be used with the functions *emis_grid* to create an emissions grid map, as shown in Fig. 8. This is helpful when the user plans to use the data to construct inputs for air quality models. The following code applies with the demo inside the VEIN model.

```
g <- make_grid(spobj = net, width = 0.00976, height = 0.00976, polygon = T)
E_CO_g <- emis_grid(spobj = net, g = g,sr = "+init=epsg:31983", type = "lines")
```

The Fig. 8 (a) shows the resulting emissions of CO in a grid with class `SpatialPolygonsDataFrame` built for a grid with resolution of 1 km representing the base year 2015. This emissions grid was built using the package **sf** (process showed in the Appendix A). Fig. 8 (b) shows the CO emissions grid of road transport from EDGAR for the same area and base year 2010 (EJ-JRC/PBL, 2016), which is the latest available year.

## 4.5   Speciation

Atmospheric simulations of ozone require knowledge about the VOC compounds and particulate matter speciation, which are necessary for solving the different chemical mechanisms. For example, a São Paulo study of ozone concentrations that used models BRAMS/SPM (Freitas et al., 2005) and WRF/Chem (Grell et al., 2005), involved detailed VOC speciation (Andrade et al., 2015). It is important to mention that there is evidence to prove that reducing Black Carbon emissions would help lower the Global Radiative Forcing and improve population health (Bond et al., 2013). Hence, the speciation of emissions is

important and VEIN provides this information. The VEIN function *speciate* splits VOC and PM into their constituents. The arguments of these functions are: emissions estimation, type of speciation, type of vehicle, fuel and Euro Standard. There are four types of PM speciation: 'bcom' (Ntziachristos and Samaras, 2016), 'tyre', 'break', and 'road' (Ntziachristos and Boulter, 2009). However, there is only one type of VOC speciation for MASP, the 'iag' (Rafee, 2015). For PM, the default speciation is 'bcom', which splits the exhaust emissions into black carbon and organic matter.

## 4.6   Input of atmospheric models

Meteorological factors influence the chemical process of pollutants in the atmosphere. Therefore, their transport and behavior in the atmosphere must be predicted by a model that includes the meteorological components ("on-line" coupling of meteorology



and chemistry), such as the Weather Research and Forecasting Chemistry model (WRF-Chem; Grell et al. 2005). This model
has been widely used around the world since its conception (2005 to 2006).

WRF-Chem, as another regional atmospheric model, requires a superficial layer of emission fluxes as input data. There are
tools to assimilate top-down emissions inventories, such as EDGAR (Olivier et al., 1996) and REanalysis of the TROpospheric
chemical composition (RETRO; Schultz 2007), using the software PREP-Chem (Freitas et al., 2011). These tools are very
important to the modeling community, however, their spatial resolutions are very limited. VEIN includes functions to generate
WRF-Chem inputs from the emissions grid with any desired resolution in the following way. VEIN estimates emissions of
different pollutants at each street and also produces emissions grids needed to do the regional modeling. This is performed
through the spatial intersection between emissions at streets and a polygon grid with the required resolution. The resulting grid
has total emissions in each grid cell proportional to the length of the streets inside each cell.

Fig. 8 shows a comparison for VEIN (Fig. 8a) and EDGAR (Fig. 8b), using emissions inventory for the CO in MASP. One
can notice that CO is spatially well represented for VEIN, by comparison with EDGAR. Furthermore, VEIN offers much more
details about the emission of this pollutant, which occurs mainly in the urban motorways due to the high volume of traffic in
these roads. The total emissions of CO emissions using VEIN are 1.73e-06 ($\mathrm{kg \cdot m^{-2} \cdot s^{-1}}$), considering the first second of a
typical Monday at 00:00 LT and EDGAR, 8.46e-08 ($\mathrm{kg \cdot m^{-2} \cdot s^{-1}}$). Therefore, VEIN estimates are 20.50 times higher than
EDGAR. This difference could be higher if compared with the morning rush hour of VEIN. However, it is important to mention
that the estimate with VEIN for this manuscript is illustrative, and that more detailed emissions inventories should be made
when comparing to others. For example, the inventory for this manuscript includes estimates only for LDV assuming that all
are PC. It does not include other types of vehicles as the total amount of vehicles were not calibrated with fuel consumption.
Ntziachristos and Samaras (2016) recommends to compare bottom-up estimates with fuel consumption in order to calibrate
inputs of emissions inventory (traffic data in this case). These differences highlight the needs for development, inter-comparison
and uncertainty evaluation of emission estimates. These results are very useful for many scientific and standardization purposes
such as health effects in air pollution studies, urban planning and strategies to cut greenhouse gas emissions.

The VEIN model provides functions to transform the emissions grids into inputs for atmospheric models. Vara-Vela et al.
presented a system for assimilating anthropogenic emissions (AAS4WRF) into the WRF-Chem model. AAS4WRF consists
of an NCL (Boulder, 2017) script that reads a text file in long format and a WRF input header for the desired domain. The
VEIN model provides the function *emis_wrf* to automatically create a data-frame in the correct format with columns longitude,
latitude, id of grid cell, pollutants, local time and GMT time in format POSIXct. The arguments of *emis_wrf* are *sdf*, which is a
list of `SpatialPolygonsDataFrames`, each per pollutant. *nr* indicates how many times the hours of estimations will be
repeated. *dmyhm* indicates the day, month, year, hour and minute of the first hour of emissions in local time. *tz* is the time-zone
and *utc* indicates the difference in hours between local and GMT time. *islist* indicates if the first argument, *df* is a list (islist
= TRUE) or a data-frame (islist = FALSE). The output of VEIN emissions are in $g \cdot h^{-1}$, and in the following code chunk,
emissions were converted to mol.

```
E_CO_g@data <- E_CO_g@data[, -1] / (12 + 16)
ldf <- list("co" = E_CO_g)
```



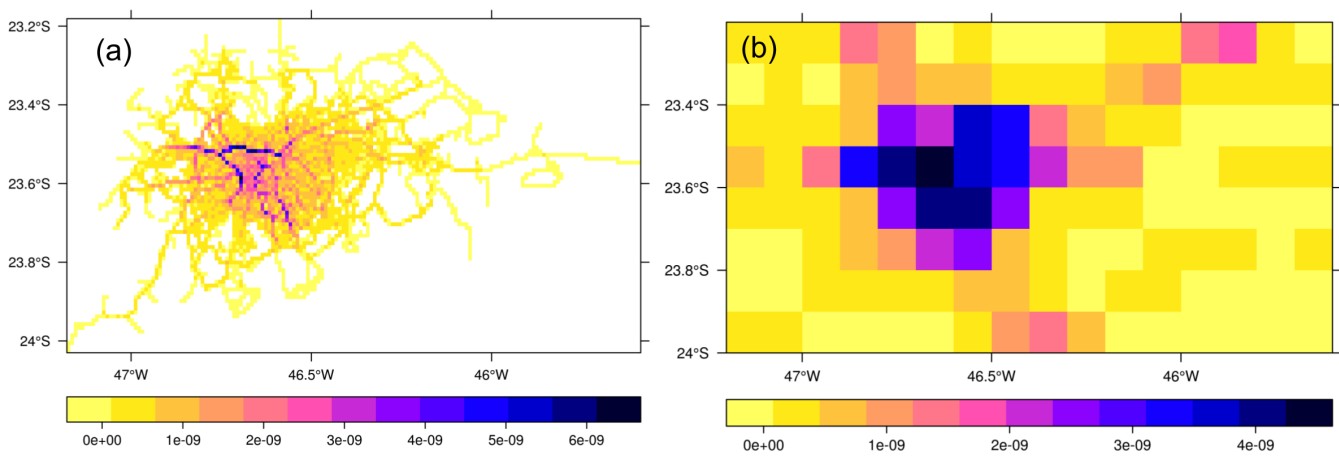

**Figure 8.** CO emissions $(\mathrm{kg \cdot m^{-2} \cdot s^{-1}})$ in MASP for (a) Monday at 00:00:00 LT, estimated with VEIN, and (b) the emissions of road transport for the same area from EDGAR.

```
df_wrf <- emis_wrf(ldf, nr = 1, dmyhm = "04-08-2014 00:00",
tz = "America/Sao_Paulo", utc = -3, islist = TRUE)
```

## 5 Discussion and Conclusions

In this manuscript we introduce the development of the Vehicular Emission INventory (VEIN model v0.2.2), an open source
model, to produce high resolution spatial and hourly emission estimation. VEIN is a tool suited for application in complex
environmental science studies, including regional atmospheric modeling. It generates inputs for air quality models in order to
forecast air pollutant concentrations or for studies of greenhouse gas emissions from vehicular sources. It can be used to study
the relationship between emissions and health effects. A recent study used VEIN estimates with a grid of 10 m resolution to
determine the relationship between vehicular emissions and birth outcomes in the western area of São Paulo (Fink et al., 2017).
VEIN can be used as a tool for urban planning in order to estimate vehicular emissions due to interventions at road networks
in most cities. It was written in an R package that includes several methods for estimating vehicular emissions in a harmonized
way.

VEIN provides functions to easily produce inputs of regional air quality models such as WRF-Chem (Grell et al., 2005)
and BRAMS/SPM (Freitas et al., 2005). In Fig. 8, the comparison between VEIN and EDGAR (EJ-JRC/PBL, 2016) of CO
emissions shows that emissions are heavily concentrated in few streets with a high volume of traffic. EDGAR emissions do
not provide this level of detail and they are lower than VEIN estimates. Furthermore, the highest spatial resolution of EDGAR
is 0.1 degree, approximately 12 km and it is possible to have better resolution with VEIN. Based on these factors, it can be
concluded that EDGAR is suitable for modeling air pollution in larger domains without considering meteorological meso-scale
interactions, including feedbacks. However, with the progressing computational advances, it would be possible to perform air



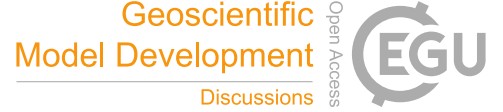

pollution modeling for larger domains with higher resolution and level of detail. VEIN can produce these necessary inputs with a bottom-up approach.

VEIN currently experiences some limitations. The first limitation is the availability of activity data. VEIN needs at least one hour of traffic data for each street considered in the estimation. This data can based on traffic simulation or traffic counts,
however, most of cities do not count with this type of data in developing countries. In this case, new data should be generated with traffic counts and interpolations. In addition, the widespread use of applications for smart-phones such as Waze (c) or Uber (c), among others, produce traffic data that eventually could be used as activity data for estimating vehicular emissions. For example, currently Google Traffic (https://developers.google.com/maps/coverage) cover several countries and this data could provide valuable information to estimate vehicular emissions in cities with non-traffic simulation or traffic counts. It is
expected that new features will be added in future versions of VEIN. One very promising feature will be the migration of the spatial dependencies into the new spatial features **sf** package. This package provides S3 classes for handling spatial data faster than its predecessor, the package **sp** (Pebesma and Bivand, 2005).

The emission factors are another aspect of VEIN that can be enhanced in future versions. They could be sourced from several emissions studies, such as tunnel studies (Pérez-Martinez et al.; Martins et al.), or others based on traffic situations whereby
emissions are sourced from driving cycles (ARTEMIS for example, André) or other experimental campaigns (Corvalán and Vargas, 2003). The International Vehicular Emissions (IVE) is a top-down vehicular emission model that has been used in different countries to estimate vehicular emissions (González et al., 2017; Wang et al., 2008). It could be possible to derive emission factors from IVE and estimate their corresponding emissions in VEIN, in order to use the capabilities of VEIN.

VEIN's purpose is to serve as a tool for air quality research and environmental management. Since air quality models need
detailed emissions species, VEIN was created with the function *speciate*. VEIN will add several new speciations into these functions, such as those in the EMEP/EEA guidelines (Ntziachristos and Samaras, 2016). In the case of Brazil, there are several studies of tropospheric ozone, which use speciation of VOC emissions as an input (Vara-Vela et al.; **?**).

*Code and data availability.* VEIN is available at the Comprehensive R Archive Network (CRAN) https://CRAN.R-project.org/package=vein and github http://www.github.com/ibarraespinosa/vein. The software version "Vehicular Emissions INventories (VEIN) v0.2.2" was devel-
oped using the R programming language. For further information about the VEIN package, contact developer Sergio Ibarra Espinosa at sergio.ibarra@usp.br. Datasets included: Emission factors *fe2015* and mileage functions *fkm* from the Environmental Agency of São Paulo (CETESB). In addition, emission and deterioration factors are provided in the form of look-up tables from Copert (Ntziachristos and Samaras, 2016). These include hot emissions factors for light duty vehicles (LDV), heavy duty vehicles (HDV), cold start, evaporative factors, cold starts, resuspension and performs speciations. Emission factor functions are included for USEPA/AP42 (EPA, 2002) estimations with
default parameters. There are also two datasets to perform the example functions, a data-frame to extrapolate morning peak hour vehicle traffic data to each hour of the week *pc_profile*, and a vehicle start pattern *pc_cold* to be used for cold start estimation. Moreover, VEIN includes a traffic simulation for the western region of São Paulo City named, *net*. Software required: VEIN depends on the package "sp" and uses some functions from "rgeos", "rgdal", "raster" and "units" packages. The user must install libraries **geos**, **gdal** and **udunits**.

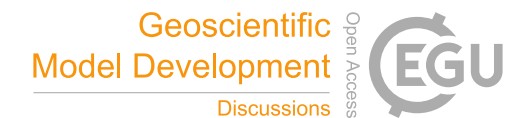



## Appendix A: Intercept grid and street emissions

```
   library(vein)
   library(sf)
   # Converting objects into "sf" and projecting to UTM zone 23S - EPSG:31983
g_sf <- st_as_sf(spTransform(g, CRSobj = "+init=epsg:31983"))
   net_sf <- st_as_sf(spTransform(net, CRSobj = "+init=epsg:31983"))
   # Calculating initial length of net_sf
   net_sf$LKM <- st_length(st_cast(net_sf[st_dimension(net_sf) == 1,]))
   # Intersecting net and grid
net_sf_g <- st_intersection(net_sf, g_sf)
   # Calculating new length of net_sf
   net_sf_g$LKM2 <- st_length(st_cast(net_sf_g[st_dimension(net_sf_g) ==1,]))
   # Checking dimensions
   dim(net_sf_g)
# Converting from sf to data.frame
   net_sf_gg <- as.data.frame(net_sf_g)
   # Eliminating column 'geometry'
   net_sf_gg <- net_sf_gg[,-171]
   # Obtaining proportional emissions of 168 hours of the week
net_sf_gg[,1:168] <- net_sf_gg[,1:168] *
   as.numeric(net_sf_gg$LKM2/net_sf_gg$LKM)
   # Agregating emissions by id of grid
   dfm <- aggregate(cbind(net_sf_gg[, 1:168]),
                   by = list(net_sf_gg$id), sum, na.rm = TRUE)
# naming columns of data-frame dfm
   names(dfm) <- c("id", paste0("h", 1:168))
   # Creating data-frame for id of grid
   gx <- data.frame(id = g$id)
   # merging data-frames
gx <- merge(gx, dfm, by="id", all.x = T)
   # Generating spatial grid and converting to "Spatial"
   E_CO_g <- as(st_sf(gx, geometry = g_sf$geometry), "Spatial")
   # Converting to WGS84 lat lon
   E_CO_g <- spTransform(E_CO_g, "+init=epsg:4326")
```





*Author contributions.* SIE designed and implemented the entire model. All authors contributed to the final manuscript.

*Competing interests.* The authors declare they have no actual or potential competing interest.

*Acknowledgements.* We thank the Environmental Agency of São Paulo State (CETESB), the programs CAPES and CNPQ (Ministry of

5  Education, Brazil) and CONICYT (Ministry of Education, Chile). These entities provided data and support that was invaluable to this study. We also thank Angel Vara-Vela from Department of Atmospheric Sciences at University of São Paulo, for discussions and tests relating to the integration between AAS4WRF and VEIN.



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
