# Peer review of "VEIN v0.2.2: an R package for bottom-up Vehicular Emissions Inventories"

_Geoscientific Model Development, 2017_

## Short Comment (SC1) · 10 Oct 2017

Sergio

The code availability section requires some clarification. v0.2.2 is not available on https://github.com/ibarraespinosa/vein/releases.

As explained in https://www.geoscientific-model-development.net/about/manuscript_types.htmlGMD is encouraging that authors make the code available at a data repository preferable with an associated DOI (digital object identifier) for the exact model version described in the paper. As your project is already on github a DOI can be easily created using Zenodo, see https://guides.github.com/activities/citable-code/ for details. Please note that in the code accessibility section you can still point the reader

to a web site for updates even if you use a DOI for a release.

I would also like to suggest that for the sake of reproducibility of results that you specify the version of R and geos, gdal and udunits you have used.

Thanks

Lutz Gross GMD Executive Editor
* * *

---

## Author Comment (AC1) · 1 Nov 2017

Dear Dr. Gross,

Thank you for your comments. The Comprehensive R Archive Network (CRAN) archives all package versions, with around 100 mirrors worldwide, and has archived the version used in the current manuscript. During the development of the manuscript, we used vein package version 0.2.2-3 available at https://cran.r-project.org/src/contrib/Archive/vein/.

We have obtained a DOI for a newer version 0.2.2-26, DOI:10.5281/zenodo.1039236 (https://zenodo.org/record/1039236) as you suggested. At the time of revising the paper, we will redo the calculations with the latest CRAN version that has a DOI. Never-

theless, all updates were minor and the results are not compromised by the updates.

In this message, we also are showing the sessionInfo() using the latest version of vein and R:

```
library(vein)
Carregando pacotes exigidos: sp
> sessionInfo()
R version 3.4.2 (2017-09-28)
Platform: x86_64-pc-linux-gnu (64-bit)
Running under: Ubuntu 17.10

Matrix products: default
BLAS: /usr/lib/x86_64-linux-gnu/blas/libblas.so.3.7.1
LAPACK: /usr/lib/x86_64-linux-gnu/lapack/liblapack.so.3.7.1

locale:
 [1] LC_CTYPE=pt_BR.UTF-8       LC_NUMERIC=C
 [3] LC_TIME=pt_BR.UTF-8        LC_COLLATE=pt_BR.UTF-8
 [5] LC_MONETARY=pt_BR.UTF-8    LC_MESSAGES=pt_BR.UTF-8
 [7] LC_PAPER=pt_BR.UTF-8       LC_NAME=C
 [9] LC_ADDRESS=C               LC_TELEPHONE=C
[11] LC_MEASUREMENT=pt_BR.UTF-8 LC_IDENTIFICATION=C

attached base packages:
[1] stats     graphics  grDevices utils     datasets
[6] methods   base

other attached packages:
```

```
[1] vein_0.2.2-26 sp_1.2-5

loaded via a namespace (and not attached):
 [1] compiler_3.4.2  rgdal_1.2-15   tools_3.4.2
 [4] units_0.4-6     yaml_2.1.14    Rcpp_0.12.13
 [7] raster_2.5-8    udunits2_0.13  grid_3.4.2
[10] rgeos_0.3-26    lattice_0.20-35
```

Also, the version of the libraries are geos v3.5.1, gdal v2.1.2 and udunits v2.2.20. The final version of the paper will include this information on the section Code Availability.

Thank you so much. On behalf of all authors,

Sergio

―――――――――――――――――――――

---

## Referee Comment (RC1) · Anonymous Referee #1 · 23 Dec 2017

This paper shows comprehensive work done as part of VEIN development. Lately, there is lot of interest in bottom-up on-road processing, so, this paper is very relevant for current context. Especially, this model focuses on emerging economies, where to obtain higher resolution and improved quality emissions is always a big challenge. However, I found some minor things which needs to be explained before accepting this paper for final publication.

Minor Comments: 1. This paper presents almost comprehensive review of emission inventories available in different parts of the world in page 2 , line 1-7. It would have been nice if they can include about how US-EPA develops their emission inventory as part of NEI.

2. The authors can mention about how their study improved methodology compared to

[Figure]

Andrade et. al., 2015 in page 2 and line 24-26.

3. Even though, the paper referred about top-down emission process studies like Ntziachristos and Sampras, 2016 and Andrade et al 2015, it did not cover the examples for bottom-up methods. Following study proposed a comprehensive methodology for bottom-up vehicular emission processing for air quality models.@article{perugu2017developing, title={Developing high-resolution urban scale heavy-duty truck emission inventory using the data-driven truck activity model output}, author={Perugu, Harikishan and Wei, Heng and Yao, Zhuo}, journal={Atmospheric Environment}, volume={155}, pages={210–230}, year={2017}, publisher={Pergamon} }. This study can be referred in page 2 , line 19.

4. Author could explain "deterioration" in page 3, line 13 when it was first time introduced. May be authors were referring vehicle deterioration in terms of emission performance.

5. F* i,j,k in the equation 3 should be explained . is it generic flow for link types of l? what kind of classification was used so that a particular link is identified that it belongs to type "l"

6. In page 3, line 19-22, it was mentioned that Capacity is found to be average of peak and free-flow speeds. But capacity of a highway link is constant throughout the day, based on their functional classification. May be authors referring traffic flow, which change hour by hour, and corresponding line has to be modified accordingly.

7. In the selection of emission factors section, the authors have discussed about vehicle type, technology and years of use etc.. I did not see important factor like fuel composition, is it inherently taken care in emission rates based on years of use?It is also looks like mostly these factors were borrowed from COPERT, I assume that model would have already taken care about it. Then, please include that clarification in this section.

8. page 5, line 20. How were vehicle deterioration factors were obtained?

9.Page 6, line 4-6, there is minor confusion about calculating cold start emissions on links. Theoretically, start emissions happen when vehicle started or if it is in idle condition after start. May be authors trying to distribute the start emissions happened at parking locations to the links, isn't it? please clarify it.

10. Page 6, line 21-22. it looks like only some seasonal days were selected in this step. You can add this as potential improvement for future versions if VEIN.

11. page 7, line 3, equation 10 : The "running " loss emissions should be by distance isn't it? Why authors considered them emissions by parks?

12. Page 11, line 7, MASP CET , is it travel demand model or micro simulation model?

13.page 11,line 12, you may use "size of" instead of weights

14.page 15, line 15, it looks alike the age vehicles were considered up to 41 years. However, the technology change in vehicles happened only 25 years before 2017. why did the researchers choose such a long time horizon as it looks the emissions from 31-41 years vehicles from figure 6(b) are very low.

15.Page 20, line 27: PM species what is bcom? and line 28 what is iag

16.page 20, line 32 , may be "in line" instead of on-line

17.page 20, line 24, missing citation for Vera-Vela et al 18. page 23, line 6-7: may be (R) instead of (c)?

In addition to above mentioned minor comments, I found some minor language issues.

Language issues: Page 2, line 17: it should be bottom-up page 2, line 33: when you first time introduce a abbreviation like VEIN, please provide the full name. page 3, line 11: it should be involved page 10, table 1 : emis_paved: It should be Re-suspension page 13, line 3: defined instead of defend page 21, line 2, may be inception instead of

conception

---

## Author Comment (AC2) · 3 Jan 2018

**Response to referees' comments**: Thank you for your comments. Please see revised draft of manuscript, included as a LaTeX supplement, with the PDF already compiled with the name vein.pdf, with changes marked in red. We included the number of each comment in the draft between parenthesis.

**Referee 1**:

**COMMENT 1**: Minor Comments. This paper presents almost comprehensive review of emission inventories available in different parts of the world in page 2 , line 1-7. It would have been nice if they can include about how US-EPA develops their emission inventory as part of NEI.

[Figure]

**REPLY 1**: We added text on page 2, lines 6-8, which shows a brief description of the NEI inventory indicating that is compiled by the US-EPA every three years and it is based on data from State, local and tribal agencies.

**COMMENT 2**: The authors can mention about how their study improved methodology compared to Andrade et. al., 2015 in page 2 and line 24-26.

**REPLY 2**: We added text on page 2, lines 34-3 and page 3, line 1. We mentioned that, despite that inventories made by Andrade et al 2015 are useful, they still suffer limitations in spatial and temporal representativeness of the emissions due to the top-down approach.

**COMMENT 3**: Even though, the paper referred about top-down emission process studies like Ntziachristos and Sampras, 2016 and Andrade et al 2015, it did not cover the examples for bottom-up methods. Following study proposed a comprehensive methodology for bottom-up vehicular emission processing for air quality models. @articleperugu2017developing, title=Developing high-resolution urban scale heavy-duty truck emission inventory using the data-driven truck activity model output, author=Perugu, Harikishan and Wei, Heng and Yao, Zhuo, journal=Atmospheric Environment, volume=155, pages=210–230, year=2017, publisher=Pergamon . This study can be referred in page 2 , line 19.

**REPLY 3**: We added text on page 2, lines 21-23. The paper suggested by the referee 1 is very appropriated because it proposes a bottom-up emission estimation method consisting in a traffic generation at link level using Spatial Regression for trucks (SPARE-Trucks) and emission rates from MOVES.

**COMMENT 4**: Author could explain "deterioration" in page 3, line 13 when it was first time introduced. May be authors were referring vehicle deterioration in terms of emission performance.

**REPLY 4**: We added text on page 6, lines 6-9. We are referring to the loss of performance of gasoline vehicles equipped with catalysts due to accumulated mileage.

**COMMENT 5**: F* i,j,k in the equation 3 should be explained . is it generic flow for link types of l? what kind of classification was used so that a particular link is identified that it belongs to type "l".

**REPLY 5**. We added text on page 4, lines 11-12. F* i,j,k was defined in equation (2) as the traffic flow. The only difference with F,i,j,k,l is that the traffic flow was extrapolated using the temporal factors for the other hours "l".

**COMMENT 6**: In page 3, line 19-22. It was mentioned that Capacity is found to be average of peak and free-flow speeds. But capacity of a highway link is constant throughout the day, based on their functional classification. May be authors referring traffic flow, which change hour by hour, and corresponding line has to be modified accordingly.

**REPLY 6**: We added text on page 4, lines 26-30. The text was misleading. The capacity is an attribute that is available in the travel demand model and this was used, with the BPR curves in equation 4, to calculate the speed at different hours. But when it is available the peak and free flow speeds, it could be calculated a simple average to obtain an average speed per link and then distribute these three speeds (peak, free flow and average), in the hours of the study.

**COMMENT 7**: In the selection of emission factors section, the authors have discussed about vehicle type, technology and years of use etc.. I did not see important factor like fuel composition, is it inherently taken care in emission rates based on years of use? It is also looks like mostly these factors were borrowed from COPERT, I assume that model would have already taken care about it. Then, please include that clarification in this section.

**REPLY 7**: We added on page 5, lines 13-16. Good observation. Fuel properties do have an effect on emission factors measured with different fuels. Previuosly we did not specify this aspect because when using emission factors based on local measurements

(not COPERT) there is no necessity of corrections. However, when using COPERT EF, corrections must be made. We included this information in the texts, suggesting the authors performs corrections in the case they use COPERT EF.

**COMMENT 8**. page 5, line 20. How were vehicle deterioration factors were obtained?

**REPLY 8**: We added text on page 6, lines 6-9. Deterioration factors comes from European emission guidelines, however, the user can use different deterioration factors.

**COMMENT 9**. Page 6, line 4-6, there is minor confusion about calculating cold start emissions on links. Theoretically, start emissions happen when vehicle started or if it is in idle condition after start. May be authors trying to distribute the start emissions happened at parking locations to the links, isn't it? please clarify it.

**REPLY 9**: We added text on page 6, lines 13-14 and 24-27. As the information about the parking location is limited, we are proposing the assignation of the emissions into the links.

**COMMENT 10**: Page 6, line 21-22. it looks like only some seasonal days were selected in this step. You can add this as potential improvement for future versions if VEIN.

**REPLY 10**: We added text on page 7, lines 27-32. Currently, the methodology of evaporative emissions into VEIN is COPERT Tier 2. We expect to include Tier 3 methods in future versions. We are also interested in investigating the methods in MOVES to include them into VEIN. In the manuscript, we proposed an alternative method to convert the emissions factors into g/km.

**COMMENT 11**: page 7, line 3, equation 10 : The "running " loss emissions should be by distance isn't it? Why authors considered them emissions by parks?

**REPLY 11**. We added text on page 6, line 21-22. The units of evaporative running losses Tier 2 is g/trip and the emission factors of hot and warm soak are g/parking. Also, in the new version of the manuscript we mentioned that if the user knows the average distance per trip, the user could transform into g/km.

**COMMENT 12**: Page 11, line 7, MASP CET , is it travel demand model or micro simulation model?

**REPLY 12**: We added text on page 12, lines 6-7. We replaced "traffic simulation" by "4-stage travel demand model"

**COMMENT 13**: page 11,line 12, you may use "size of" instead of weights.

**REPLY 13**: Done.

**COMMENT 14**: page 15, line 15, it looks alike the age vehicles were considered up to 41 years. However, the technology change in vehicles happened only 25 years before 2017. why did the researchers choose such a long time horizon as it looks the emissions from 31-41 years vehicles from figure 6(b) are very low.

**REPLY 14**: We added text on page 15, lines 31-33 and page 16, line 1. We considered a distribution of vehicles up to 41 years of use because it is more representative of the vehicles in circulation. VEIN can perform these calculations very quickly. This feature allows using VEIN as a tool for scrapping policies in order to accomplish emission targets.

**COMMENT 15**: .Page 20, line 27: PM species what is bcom? and line 28 what is iag.

**REPLY 15**: We added text on page 2, lines 25 and 26-28. 'bcom' is the name in the function 'speciate' for splitting Black Carbon and Organic Matter. 'iag' comes from Institue of Astronomy, Geophysics and Atmospheric Sciences from the University of São Paulo, because most of the speciation and air quality modeling in Brazil comes from this institute and the speciation 'iag' is based on measurements made by researchers of this institute.

**COMMENT 16**: page 20, line 32 , may be "in line" instead of on-line. **REPLY 16**: Done.

**COMMENT 17**: page 20, line 24, missing citation for Vera-Vela et al 18. page 23, line 6-7: may be (R) instead of (c)?

**REPLY 17**: DONE. We also added pages 3-6 to indicate that we are developing another model named 'eixport' that will produce inputs not only for WRF-Chem, but also for BRAMS-SPM (Freitas et al., 2005), R-LINE (Snyder et al., 2013) and more.

**Language issues**: Page 2, line 17: it should be bottom-up. DONE.

page 2, line 33: when you first time introduce a abbreviation like VEIN, please provide the full name. DONE.

page 3, line 11: it should be involved. DONE.

page 10, table 1 : $emis\_paved$: It should be Re-suspension. DONE.

page 13, line 3: defined instead of defend. DONE.

page 21, line 2, may be inception instead of conception. DONE.

Please also note the supplement to this comment:
https://www.geosci-model-dev-discuss.net/gmd-2017-193/gmd-2017-193-AC2-supplement.zip

―――――――――――――――

---

## Referee Comment (RC2) · Anonymous Referee #2 · 26 Apr 2018

This paper presents a very valuable contribution to the development of emission inventories of mobile sources with the necessary resolution to run a chemical transport model at a regional scale.

This issue is of growing interest in developing countries, where the control of air quality becomes increasingly critical. The model is even more valuable because it is developed in a language of free availability. It is based on the model used in Europe (COPERT), and also uses data measured in Brazil. This fact of course adds value but, on the other hand given the special fuel mix used in Brazil, some of these values (particularly those corresponding to organic compounds) may be biased. In any case, in general terms, the program presents flexibility for the use of other data, and for that reason its use in other regions is possible. For all the above, I suggest accepting this work for publication

after some minor corrections detailed below.

1. The relationship between hourly average speed and traffic flow is given by equation 4, which includes the parameters alfa and beta. The authors suggest default values but allow the users of the model to choose local data. Perhaps with a better explanation about the variable Capacity this issue will be clarify but, in any case, the explanation on how these default values were established is needed, and to what extent they depend on the local type of fleet / circulation circumstances.

2. In the Emission Factors options explained in item 2.2, it is not clear the difference between the option 2) Emission factors from local sources (line 31, page 4) and the $EFlocal_{j,k,m}$ that represents the constant emission factor (not speed functions). If they represent the same (i.e. the EFs measured in any place, on the basis of dynamometer's experiments) the authors should explain why the option 2) is given. In the case to have other experiments, such as on-road measurements or tunnel studies, how these numbers are included? This item has to be expanded in order to give more details about the use of local EFs data.

3. The authors does not mention the difference in the emissions regarding the type of fuel used, which is particularly relevant for example in Brazil where almost all new cars sold can run on any combination of gasoline and ethanol. This deserve clarifications in several parts of the manuscript:

3.a. Deterioration factors: The model includes an emission factor database (fe2015) that does not include this factor. Nevertheless the authors report the use of deterioration factors from Ntziachristos and Samaras (2016), who said that these factors should not be used to provide the deterioration of emissions where an older fuel is used in a newer technology (e.g. use of Fuel 2000 in Euro 4 vehicles) and, therefore, cannot be used for other type of fuels. This reviewer agrees that it is necessary to make some consideration in this regard, and that the availability of these numbers is scarce, so that considering European values (due to the lack of better ones) is a valid option. However, it is necessary to prevent end users of the model about this assumption including a comment in the manuscript because it is known that the use of ethanol accelerates the deterioration of the engine.

3.b. Speciation schemes –dependence with fuel composition. An unknown set of schemes from Rafee (2015), whose reference is in Portuguese, has been considered. It is necessary to include a brief explanation about this work including, for example, which type of fuels these schemes include.

4. The data of vehicle start pattern taken from the IVE experience in Brazil, how can be extrapolated to other cities/regions? Please, clarify the variability of this parameter, and its dependence on the type of technologies used in the different countries.

5. Minor comments 5.a. : Page 5 line 28 " studies report that when ambient temperature is -7C, emissions are one order of magnitude higher than at 22C (Ludykar et al., 1999)", please clarify the corresponding pollutant (the numbers are totally different between compounds).

5.b. In Page 4 line 25 the authors define the meaning of Emission factor, referring to Tinus Pulles definition, but for the manuscript define "an emission factor is the mass of pollutant emitted by the vehicular type, technology and years of use", but the activity data was not included in the definition (km travelled). 5.c. Page 5 lines 8 to 11: Mileage driven with a cold engine/catalyst: the authors said that they included in the model the cold starts recorded during the implementation of the International Vehicle10 Emissions (IVE) model, but is not enough clear if the data from COPERT are also included. Please, also clarify if other data about this parameter may be included by other modelers.

5.d. Page 6 line 5: "This is an important aspect that will be reviewed in future versions of VEIN". Please, clarify the intention.

5.e. It is not clear the input data needed to estimate the daily cicle, (1) rush hours

during the morning pick or (2) the available data in each country/city, combining with a TF matrix accordingly, with a value of 1 for the maximum flow, at the time that it occurs.

5.f. Please, clarify if it would be possible to include other speciation schemes

---

## Editor Comment (EC1) · A. Colette (Editor) · 2 May 2018

Review of VEIN v0.2.2: an R package for bottom-up Vehicular Emissions Inventories submitted to GMD by Sergio Ibarra-Espinosa, Rita Ynoue, Shane O'Sullivan, Edzer Pebesma, María de Fátima Andrade, and Mauricio Osses.

The paper presents in detail a new model to assess air pollutant and greeenhouse gases emissions due to road transportation. The model offers interesting capacities to make use of local information about traffic counts in conjunction to well established methodologies to model emission fluxes. The methodology is well explained and the model itself is available as an open source R package, therefore offering interesting perspectives for users and further development.

[Figure]

**GMDD**

General comment

The model builds upon well established methodologies, in particular those of the COP-ERT model. More details about the added value of VEIN (for instance in terms of mapping and use of local bottom up information) should be provided in the introduction to allow the reader to better understand the complementarity compared to COPERT or other traffic models. Since VEIN relies heavily on COPERT data and methodologies, a more explicit acknowledgment is needed indicating if such input data are available for public use and/or if the support from COPERT developpers was granted.

Specific comment

Why is traffic count data only available for the morning rush hours (P4L1) if they are obtained from automated stations (P4L10) ?

P6L6 if monthly average temperature is already taken into account as recommended in the COPERT methodology, how will it be improved in the future versions of VEIN ?

Sections 2.4 and 4.5 : more details should be added on the chemical speciation of emissions.

P14L22 : why is the green line so different from the other two in terms of variability ? is it obtained from actual data instead of a statistical fit ?

P21L15 : a ratio 20.5 between estimates with VEIN and EDGAR calls for further justification. Is it for the whole MASP area or focusing only on major motorways ? A comparison of annual total emissions should be provided.

Technical Comments

- P1L7 : define Âń factors Âż

- P1L16 : add greeenhouse gases

- P2L28 : define HC, how do they differ from VOC mentionned later in the manuscript

?

- P4L28 : which is the computer programme being referred to ?

- P8L17 : the colors can not be seen in Fig 1.

- P13L3 : Âń defined Âż instead of Âń defend Âż

- P21L3 : replace Âń another Âż by Âń other Âż

- P23 L14, L15 : years are missing from the references

- P23L23 : the purpose of the last sentence is unclear.

---

## Author Comment (AC4) · 3 May 2018

**Response to editor** Thank you for your comments. Please see our revised draft of the manuscript, in the attached file pdf, with changes marked in brown. The pdf also includes replies for Referee 1 in red and replies for Referee 2 in blue.

**General Comment**: The model builds upon well established methodologies, in particular those of the COPERT model. More details about the added value of VEIN (for instance in terms of mapping and use of local bottom up information) should be provided in the introduction to allow the reader to better understand the complementarity compared to COPERT or other traffic models. Since VEIN relies heavily on COPERT data and methodologies, a more explicit acknowledgment is needed indicating if such

input data are available for public use and/or if the support from COPERT developpers was granted.

**Reply**: We agree with you. We added the lines 5-7 and 9-10 on page 3 showing that the emissions factors of speed functions VEIN are based on Ntziachristos and Samaras (2016) and other authors. VEIN is capable of representing spatial objects. We also included acknowledgments on lines 18-20 on page 28.

**Comment 1**: Why is traffic count data only available for the morning rush hours (P4L1) if they are obtained from automated stations (P4L10)

**Reply 1**: Red and blue lines in the same paragraph on page 4 expand further on the explanation between morning traffic data and traffic counts. Basically, 4-stage travel demand models produce outputs with traffic at **each street** for peak hours. This data can be used in VEIN, the user needs to extrapolate the traffic on street to the other hours of the day, and also, days of the week. This role is played by the $TF$ matrices play, to expand morning rush hour traffic on the streets. The data needed to build $TF$ can be obtained from automatic/manual traffic count stations, which deliver the counts in certain points on space.

**Comment 2**: P6L6 if monthly average temperature is already taken into account as recommended in the COPERT methodology, how will it be improved in the future versions of VEIN ?.

**Reply 2**: We removed that line.

**Comment 3**: Sections 2.4 and 4.5 : more details should be added on the chemical speciation of emissions.

**Reply 3**: We added more information and the other Reviewers also made this same request. The added information is on Page 8 lines 28-32, page 9 lines 1-3, page 22 lines 25-34 and page 23 lines 1-3.

**Comment 4**: P14L22 : why is the green line so different from the other two in terms of

variability ? is it obtained from actual data instead of a statistical fit ?

**Reply 4**: The lines 3-5 on page 15 show that the distribution of vehicles by age of use can also be obtained using the function $my\_age$ results in green line on Fig. 5a. As mentioned, this data is based on fuel sales and is sourced from the Environment agency of SãøPaulo CETESB.

**Comment 5**: P21L15 : a ratio 20.5 between estimates with VEIN and EDGAR calls for further justification. Is it for the whole MASP area or focusing only on major motorways ? A comparison of annual total emissions should be provided.

**Reply 5**: The 20.5 comes from the comparison between the sum of CO emissions due to transport for the same area with VEIN at 00:00 (local time) and EDGAR, covering the whole MASP area. However, on page 23 lines 25-29 we mention that an emissions inventory needs to be compared and calibrated with fuel consumption for that area. We also mention that the estimate in the manuscript is illustrative and focussed on VEIN capabilities. Nevertheless, this highlights the need for inter-comparison between inventories.

**Technical comments, 1**: P1L7 : define ?? factors ??.

**Reply Technical comments, 1**: We did not understand.

**Technical comments, 2**: P1L16 : add greeenhouse gases.

**Reply Technical comments, 2**: On page 1, lines 19-20 we mention the greenhouse gases inventories.

**Technical comments, 3**: P2L28 : define HC, how do they differ from VOC mentionned later in the manuscript

**Reply Technical comments, 3**:Great observation. Volatile Organic Compounds (VOC), now changed on the text. We are referring to hydrocarbon without methane.

**Technical comments, 4**: P4L28 : which is the computer programme being referred to

[Figure]

?

**Reply Technical comments, 4**: Computer programme to calculate emissions from road transport means COPERT. We changed to COmputer Programme to calculate Emissions from Road Transport on page 5, line 18.

**Technical comments, 5**: P8L17 : the colors can not be seen in Fig 1.

**Reply Technical comments, 5**: Great observation. Initially that diagram had colors, but later it became black and white. Now it is fixed.

**Technical comments, 6**: P13L3 : ?? defined ?? instead of ?? defend ??

**Reply Technical comments, 6**: We updated that phrase in page 13, line 33 and page 14, lines 1-3.

**Technical comments, 8**: P21L3 : replace ?? another ?? by ?? other ??

**Reply Technical comments, 8**: P21L3 : We edited the phrase about WRF-Chem and emissions on 9, page 23.

**Technical comments, 9**: P23 L14, L15 : years are missing from the references.

**Reply Technical comments, 9**: Thanks. Now it is fixed.

**Technical comments, 10**: P23L23 : the purpose of the last sentence is unclear.

**Reply Technical comments, 10**: That page and line belongs to the *Code Availability* section on the origin manuscript and there is no explanation for its lack of clarity.
* * *
**VEIN v0.2.2: an R package for bottom-up Vehicular Emissions Inventories**

Sergio Ibarra-Espinosa[1], Rita Ynoue[1], Shane O'Sullivan[2], Edzer Pebesma[3], Maria de Fátima Andrade[1], and Mauricio Osses[4]

[1]Department of Atmospheric Sciences, Universidade de São Paulo, Rua do Matão 1226, São Paulo, SP, Brazil
[2]Department of Pathology, Faculdade de Medicina, Universidade de São Paulo, Av. Dr. Arnaldo 455, São Paulo, SP, Brazil
[3]Institute for Geoinformatics, Westfälische Wilhelms-Universität Münster, Heisenbergstraße 2, 48149 Münster, Germany
[4]Department of Mechanical Engineering, Universidad Técnica Federico Santa María, Vicuña Mackenna 3939, Santiago, Chile

*Correspondence to:* Sergio Ibarra-Espinosa (zergioibarra@gmail.com)

**Abstract.**

Emission inventories are the quantification of pollutants from different sources. They provide important information not only for climate and weather studies, but also for urban planning and environmental health protection. We developed an open source model (named VEIN v0.2.2) that provides high resolution vehicular emissions inventories for different fields of studies. We focused on vehicular sources at street and hourly levels due to the current lack of information about these sources, mainly in developing countries. The type of emissions covered by VEIN are: exhaust (hot and cold) and evaporative considering the deterioration of the factors. VEIN also performs speciation and incorporates functions to generate and spatially allocate emissions databases. It allows users to load their own emissions factors, but it also provides emissions factors from the road transport model (Copert), the United States Environmental Protection Agency (EPA) and Brazilian databases. The VEIN model reads, distributes by age of use and extrapolates hourly traffic data, and estimates hourly and spatially emissions. Based on our knowledge, VEIN is the first bottom-up vehicle emissions software that allows input to the WRF-Chem model. Therefore, the VEIN model provides an important, easy and fast way of elaborating or analyzing vehicular emissions inventories, under different scenarios. The VEIN results can be used as an input for atmospheric models, health studies, air quality standardizations and decision making.

**1 Introduction**

Emissions inventory is a quantification of pollutants discharged into the atmosphere by different sources (Pulles and Heslinga, 2010). This quantification is vital for regulatory and scientific purposes, because it allows to monitor the state of the Earth's atmosphere and climate. It also allows to create air quality standards, which will protect ecosystems and human health. For instance, the Intergovernmental Panel on Climate Change (IPCC) includes a dedicated task force, separated from the other three working groups, only for the purpose of greenhouse gas emissions inventory issues (Paustian et al., 2006).

In this instance, there are several emissions inventories that use different input data and approaches for different scales. One of the most frequently used inventories is the Emission Database for Global Atmospheric Research (EDGAR; Olivier

**Fig. 1.**

---

## Author Comment (AC5) · 3 May 2018

**Response to comments of Referee 2:** Thank you for your comments. Please see our revised draft of the manuscript, in the attached file vein3.pdf, with changes marked in blue with the number representing each of your comments between parentheses. This manuscript also includes the replies for Referee 1 in red and some replies for editor in brown.

**Comment 1**: The relationship between hourly average speed and traffic flow is given by equation 4, which includes the parameters alfa and beta. The authors suggest default values but allow the users of the model to choose local data. Perhaps with a

better explanation about the variable Capacity this issue will be clarify but, in any case, the explanation on how these default values were established is needed, and to what extent they depend on the local type of fleet / circulation circumstances.

**Reply 1**: We added descriptions between line 33-34 of page 4 and line 1-7 of page 5. We added cited definitions of capacity from the Highway Capacity Manual. We also cited manuscripts of authors that fitted the BPR parameters alpha and beta based on local recordings of speed and traffic to represent the local characteristics of traffic fleet and circulation.

**Comment 2**: In the Emission Factors options explained in item 2.2, it is not clear the difference between the option 2) Emission factors from local sources (line 31, page 4) and the $EFlocal_{j,k,m}$ that represents the constant emission factor (not speed functions). If they represent the same (i.e. the EFs measured in any place, on the basis of dynamometer?s experiments) the authors should explain why the option 2) is given. In the case to have other experiments, such as on-road measurements or tunnel studies, how these numbers are included? This item has to be expanded in order to give more details about the use of local EFs data..

**Reply 2**: We added descriptions between lines 22-26 of page 5 for item 2 relating to local emission factors. We also added the Appendix B, which shows examples about how to estimate emissions with VEIN using COPERT, local and scaled emission factors.

**Comment 3**: The authors does not mention the difference in the emissions regarding the type of fuel used, which is particularly relevant for example in Brazil where almost all new cars sold can run on any combination of gasoline and ethanol. This deserve clarifications in several parts of the manuscript:

*a) Deterioration factors:* The model includes an emission factor database (fe2015) that does not include this factor. Nevertheless the authors report the use of deterioration factors from Ntziachristos and Samaras (2016), who said that these factors should not be used to provide the deterioration of emissions where an older fuel is used in a newer technology (e.g. use of Fuel 2000 in Euro 4 vehicles) and, therefore, cannot be used for other type of fuels. This reviewer agrees that it is necessary to make some consideration in this regard, and that the availability of these numbers is scarce, so that considering European values (due to the lack of better ones) is a valid option. However, it is necessary to prevent end users of the model about this assumption including a comment in the manuscript because it is known that the use of ethanol accelerates the deterioration of the engine

*b) Speciation* schemes dependence with fuel composition. An unknown set of schemes from Rafee (2015), whose reference is in Portuguese, has been considered. It is necessary to include a brief explanation about this work including, for example, which type of fuels these ses include..

**Reply 3a)**: Great obvervation. Yes Ntziachristos and Samaras (2016) show that new technologies associated with newer standards need fuel with proper quality. For instance, as on page 52 of Ntziachristos and Samaras (2016, https://www.eea.europa.eu/publications/emep-eea-guidebook-2016/part-b-sectoral-guidance-chapters/1-energy/1-a-combustion/1-a-3-b-i/view), the subsection on **Fuel effects**, quoting: "emission standards of Euro 3 technology (introduced 2000) are achieved with Fuel 2000, and the more stringent emission standards of Euro 4 and 5 with Fuel 2005". This refers to the effect that, older vehicles started to consume the new and better fuel, which also led to diminishing emissions also in these types of vehicles. To take into account this effect, the Equation (24):

$$FCeHOT_{i,k,r} = FCor_r i, k, Fuel/FCorr_{i,k,Base} * eHOTi, k, r$$

cannot be used to account for deterioration. This means that

$$FCorr_{Fuel}/FCorr_{Base}$$

is restricted to values below or equal to 1. Currently, there are no fuel correction functions in VEIN on CRAN repository. However, we didincluded the function $fuel_corr$ on GitHub repository.

On the other hand, the section about **Emission degradation due to vehicle age** on page 51, presents the deterioration factors depending on accumulated mileage and technology of vehicles with 3-way catalizers. To our knowledge, there are not available deterioration factors for vehicles consuming bio-fuels. Hence, we agree that, in the absence of better data, using COPERT deterioration factors in a Brazilian fleet is valid. We included the information between lines 24-30 of page 16 to warn users of COPERT factors, by mentioning that users must include deterioration and fuel effects factors.

**Reply 3 b)**: The reference of Rafee (2015) was replaced with an Ph.D thesis in english of the author of this manuscript Ibarra S. (2017), which is included in the references. The new speciation updates the exhaust speciation of NMHC which is based on other studies. We added the information in between lines 28-32 of page 8 and pages 1-3 of page 9.

**Comment 4**: The data of vehicle start pattern taken from the IVE experience in Brazil, how can be extrapolated to other cities/regions? Please, clarify the variability of this parameter, and its dependence on the type of technologies used in the different countries.

**Reply 4**: The data of start patterns should be generated by users by using local data. For instance Gonzalez et al (2017) generated a start pattern using surveys to estimate vehicular emissions using IVE. However, the user eventually could use the start pattern

available in VEIN only in the absence of other data. This pattern is $\beta$ parameter by type of vehicle on equation (7). The function $ef_l dv_c old$ covers this type of emissions, based on Ntziachristos and Samaras (2016), which depends on the type of vehicle, temperature, size of engine, fuel, euro standard and pollutant. As consequence, this function can be used in other countries by identifying the equivalence between local and Euro standard, as shown on Table 2 for the Brazilian case. We added explanations on page 7, between lines 19-21.

**Minor comments 5.a.** : Page 5 line 28 ? studies report that when ambient temperature is -7C, emissions are one order of magnitude higher than at 22C (Ludykar et al., 1999)?, please clarify the corresponding pollutant (the numbers are totally different between compounds).

**Reply 5a**: Great observation. We specified the change in each pollutant by citing Table 1 of the same study between lines 4-7 of page 7.

**Minor comments 5.b**: In Page 4 line 25 the authors define the meaning of Emission factor, referring to Tinus Pulles definition, but for the manuscript define ?an emission factor is the mass of pollutant emitted by the vehicular type, technology and years of use?, but the activity data was not included in the definition (km travelled).

**Reply 5b**: In Page 5, line 16, we added the phrase: "by traveled distance, as mass of pollutant / distance", representing the travelled distance of the activity.

**Minor comments 5.c**: Page 5 lines 8 to 11: Mileage driven with a cold engine/catalyst: the authors said that they included in the model the cold starts recorded during the implementation of the International Vehicle Emissions (IVE) model, but is not enough clear if the data from COPERT are also included. Please, also clarify if other data about this parameter may be included by other modelers.

**Reply 5.c**: Equation 7 shows the approach for estimating cold start emissions. The incorporation of the start pattern from the IVE study was meant to bse used as the $\beta$,

the fraction of mileage driven with cold engine/catalyst. This approximation represents the percentage of distance driven under cold conditions. The user could also follow Ntziachristos and Samaras (2016) approach for estimating $\beta = 0.6474 - 0.02545 * lengthtrip - (0.00974 - 0.000385 * lenthtrip) * ta$. We added an explanation between lines 21-23 on page 7.

**Minor comments 5.d**: 5.d. Page 6 line 5: ?This is an important aspect that will be reviewed in future versions of VEIN?. Please, clarify the intention.

**Reply 5.d**: We removed that line which you have highlighted. We intended to imply that start patterns used in $\beta$ would need to be tested, but this has already been tested in IVE inventories.

**Minor comments 5.e**:It is not clear the input data needed to estimate the daily cycle, (1) rush hours during the morning pick or (2) the available data in each country/city, combining with a TF matrix accordingly, with a value of 1 for the maximum flow, at the time that it occurs.

**Reply 5.e.**:We added further explanations on page 4, lines 16-18, 21-22.

**Minor comments 5.f**:Please, clarify if it would be possible to include other speciation schemes.

**Reply 5.f**: Very important comment. Currently, the chemical mechanisms in VEIN includes the COV speciation 'iag' which splits the VOC emissions for the mechanism Carbon Bond Mechanism Z. If the user intends to use other mechanisms, needs to know how to speciate the COV and PM based on the user's own data. This means that the user must know the percentages to split the pollutants. In this case, the user could use that percentage in the argument $k$ for any VEIN function of emission factor for the respective type of vehicle. The user could then use this to estimate the emissions for that fraction of vehicles. For example, if the user knows that 5% of COV emissions of LDVs consuming diesel are Xylenes, then the

user must use the function $ef\_ldv\_speed$ or $ef\_ldv\_scaled$ (or its own local emission factors) with the argument k = $5/100$. The argument k is just a factor added to the resulting emission function. Finally, the user must aggregate the emissions by pollutant. We added this explanation on page 22, lines 29-33 and and page 23, lines 1-3. Moreover, users can contribute to the development of the model with the pull requests on GitHub repository https://github.com/atmoschem/vein (previously https://github.com/ibarraespinosa/vein).
* * *
[Figure]

**VEIN v0.2.2: an R package for bottom-up Vehicular Emissions Inventories**

Sergio Ibarra-Espinosa[1], Rita Ynoue[1], Shane O'Sullivan[2], Edzer Pebesma[3], Maria de Fátima Andrade[1], and Mauricio Osses[4]

[1]Department of Atmospheric Sciences, Universidade de São Paulo, Rua do Matão 1226, São Paulo, SP, Brazil
[2]Department of Pathology, Faculdade de Medicina, Universidade de São Paulo, Av. Dr. Arnaldo 455, São Paulo, SP, Brazil
[3]Institute for Geoinformatics, Westfälische Wilhelms-Universität Münster, Heisenbergstraße 2, 48149 Münster, Germany
[4]Department of Mechanical Engineering, Universidad Técnica Federico Santa María, Vicuña Mackenna 3939, Santiago, Chile

*Correspondence to:* Sergio Ibarra-Espinosa (zergioibarra@gmail.com)

**Abstract.**

Emission inventories are the quantification of pollutants from different sources. They provide important information not only for climate and weather studies, but also for urban planning and environmental health protection. We developed an open source model (named VEIN v0.2.2) that provides high resolution vehicular emissions inventories for different fields of studies. We focused on vehicular sources at street and hourly levels due to the current lack of information about these sources, mainly in developing countries. The type of emissions covered by VEIN are: exhaust (hot and cold) and evaporative considering the deterioration of the factors. VEIN also performs speciation and incorporates functions to generate and spatially allocate emissions databases. It allows users to load their own emissions factors, but it also provides emissions factors from the road transport model (Copert), the United States Environmental Protection Agency (EPA) and Brazilian databases. The VEIN model reads, distributes by age of use and extrapolates hourly traffic data, and estimates hourly and spatially emissions. Based on our knowledge, VEIN is the first bottom-up vehicle emissions software that allows input to the WRF-Chem model. Therefore, the VEIN model provides an important, easy and fast way of elaborating or analyzing vehicular emissions inventories, under different scenarios. The VEIN results can be used as an input for atmospheric models, health studies, air quality standardizations and decision making.

**1 Introduction**

Emissions inventory is a quantification of pollutants discharged into the atmosphere by different sources (Pulles and Heslinga, 2010). This quantification is vital for regulatory and scientific purposes, because it allows to monitor the state of the Earth's atmosphere and climate. It also allows to create air quality standards, which will protect ecosystems and human health. For instance, the Intergovernmental Panel on Climate Change (IPCC) includes a dedicated task force, separated from the other three working groups, only for the purpose of greenhouse gas emissions inventory issues (Paustian et al., 2006).

In this instance, there are several emissions inventories that use different input data and approaches for different scales. One of the most frequently used inventories is the Emission Database for Global Atmospheric Research (EDGAR; Olivier

**Fig. 1.**